# *GH3* Gene Family Identification in Chinese White Pear (*Pyrus bretschneideri*) and the Functional Analysis of *PbrGH3.5* in Fe Deficiency Responses in Tomato

**DOI:** 10.3390/ijms252312980

**Published:** 2024-12-03

**Authors:** Pengfei Wei, Guoling Guo, Taijing Shen, Anran Luo, Qin Wu, Shanshan Zhou, Xiaomei Tang, Lun Liu, Zhenfeng Ye, Liwu Zhu, Bing Jia

**Affiliations:** 1School of Horticulture, Anhui Agricultural University, Hefei 230036, China; weipengfeiaau@163.com (P.W.); ggl1995@ahau.edu.cn (G.G.); 21720259@stu.ahau.edu.cn (T.S.); 15682231660@163.com (A.L.); tangxiaomei@ahau.edu.cn (X.T.); liulun2016@163.com (L.L.); yezhenfeng@ahau.edu.cn (Z.Y.); 2Jinzhai Modern Agricultural Cooperation Center, Integrated Experimental Station in Dabie Mountains, Anhui Agricultural University, Lu’an 237000, China; 18256413722@163.com (Q.W.); zss13856984636@163.com (S.Z.)

**Keywords:** Fe deficiency hypersensitivity, genome-wide analysis, IAA, pear, *PbrGH3s*, reactive oxygen species

## Abstract

Iron (Fe) deficiency poses a major threat to pear (*Pyrus* spp.) fruit yield and quality. The *Gretchen Hagen 3* (*GH3*) plays a vital part in plant stress responses. However, the *GH3* gene family is yet to be characterized, and little focus has been given to the function of the *GH3* gene in Fe deficiency responses. Here, we identified 15 GH3 proteins from the proteome of Chinese white pear (*Pyrus bretschneideri*) and analyzed their features using bioinformatics approaches. Structure domain and motif analyses showed that these PbrGH3s were relatively conserved, and phylogenetic investigation displayed that they were clustered into two groups (GH3 I and GH3 II). Meanwhile, *cis*-acting regulatory element searches of the corresponding promoters revealed that these *PbrGH3s* might be involved in ABA- and drought-mediated responses. Moreover, the analysis of gene expression patterns exhibited that most of the *PbrGH3s* were highly expressed in the calyxes, ovaries, and stems of pear plants, and some genes were significantly differentially expressed in normal and Fe-deficient pear leaves, especially for *PbrGH3.5*. Subsequently, the sequence of *PbrGH3.5* was isolated from the pear, and the transgenic tomato plants with *PbrGH3.5* overexpression (OE) were generated to investigate its role in Fe deficiency responses. It was found that the OE plants were more sensitive to Fe deficiency stress. Compared with wild-type (WT) plants, the rhizosphere acidification and ferric reductase activities were markedly weakened, and the capacity to scavenge reactive oxygen species was prominently impaired in OE plants under Fe starvation conditions. Moreover, the expressions of Fe-acquisition-associated genes, such as *SlAHA4*, *SlFRO1*, *SlIRT1*, and *SlFER*, were all greatly repressed in OE leaves under Fe depravation stress, and the free IAA level was dramatically reduced, while the conjugated IAA contents were notably escalated. Combined, our findings suggest that pear *PbrGH3.5* negatively regulates Fe deficiency responses in tomato plants, and might help enrich the molecular basis of Fe deficiency responses in woody plants.

## 1. Introduction

Pear (*Pyrus* spp.) is the third most highly appreciated horticultural crop with nutritional and economic benefits in China [1]. The growth and development of pear trees, as well as the fruit production and quality, are greatly associated with microelements, such as iron (Fe) [2]. Fe is a fundamental micronutrient for plant normal growth and development [3], and acts as a vital cofactor in chlorophyll biosynthesis, energy metabolism, photosynthesis, antioxidation, and respiration [4]. Although Fe is abundant in soil, the bioavailable Fe for plants is poor due to its low solubility in saline–alkaline and calcareous soils [5]. Therefore, Fe deficiency has become a widespread concern in agricultural production, reducing the yield and quality of pear plants [6].

To cope with Fe-limitation, plants have evolved two primary strategies, distinguished as strategy I for dicots and non-graminaceous monocots, and strategy II for graminaceous plants [7]. Pear trees employ the strategy I system to utilize Fe from the soil, where rhizosphere acidification determined by plasma membrane proton ATPase 2 (AHA2) is a decisive step to improve Fe dissolution. Hereafter, the Fe^3+^ ions are reduced to Fe^2+^ ions in the presence of ferric reductase oxidase 2 (FRO2), and the resulting Fe^2+^ ions are subsequently introduced into roots via the metal transporter iron-regulated transporter 1 (IRT1) [8,9,10]. Therefore, an effective behavior to elevate the expressions of these Fe-uptake-related genes is essential for Fe absorption in Fe-limited situations.

Transcriptional regulation is a pivotal way to regulate gene expressions [11], and plants activate a variety of transcription factors (TFs) that act upstream of *IRT1*, *AHA2*, and *FRO2* to express under Fe deficiency stress [12,13]; among them, the basic helix–loop–helix (bHLH) family is indispensable, mainly comprising bHLH29 (FIT), bHLH18/19/20/25, bHLH47/121, bHLH34/104/105/115, and bHLH38/39/100/101 [14,15,16]. In addition, Fe-deficiency-stressed plants can adjust the endogenous hormone level, and the altered hormone profile, in turn, triggers numerous operative Fe deficiency responses (FDR) to further help them resist Fe-deprived stress [17,18,19], in which auxin (IAA) plays an important part. It was found that exogenous gamma-aminobutyric acid (GABA) can promote the accumulation of IAA in cucumber under Fe starvation conditions, and the boosted IAA level would assist in up-regulating the expressions of *CsHA1*, *CsFRO2*, and *CsIRT1* [20]. In *Arabidopsis*, endogenous IAA can positively regulate the abundance, transport, activities, and phosphorylation of AHA1/2/11, thus promoting the H^+^ efflux, which is urgently required for plants to withstand Fe scarcity [21,22]. Meanwhile, exogenous IAA can also facilitate root development to stimulate polar IAA transport from shoots to roots, thus alleviating Fe deficiency stress in rice mutant *ospin1b* [23]. Therefore, given that the IAA function is inseparable from its signaling, the corresponding responders in this pathway might play crucial roles in the FDR of plants.

The *Gretchen Hagen 3* (*GH3*) gene is one of the essential members in IAA early responsiveness, and functions in the conjugation of IAA to various amino acids, including leucine, alanine, aspartate, glutamate, and phenylalanine [24,25,26]. Since it was first recognized in soybean [27], the *GH3* gene family has been successively discovered in *Arabidopsis thaliana* [24], *Oryza sativa* [28], *Malus domestica* [29], *Zea mays* [30], *Gossypium arboretum* [31], *Triticum aestivum* [32], *Ipomoea batatas* [33], *Camellia sinensis* [34], etc. Therefore, the function of *GH3* has been widely explored, including regulatory roles in lateral root initiation and development, leaf expansion, and pathogen defense [35,36,37,38]. Beyond these, mounting studies have proved that *GH3s* participate in regulating the tolerance of plants to plenty of abiotic stresses, and the corresponding responses are largely achieved in a reactive oxygen species (ROS)-dependent manner [39,40,41]. However, whether the *GH3* gene is relevant to FDR remains mysterious.

ROS are predominantly present in the form of peroxide (H_2_O_2_) and superoxide radical (O_2_^−^) in plants, and serve as signaling molecules with dual effects in regulating multiple biological processes [42]. Mounting evidence has shown that Fe deficiency is closely associated with ROS. Fe deficiency would trigger the over-production of ROS and cause irretrievable membrane damage to plants, hence leading to Fe deficiency hypersensitivity [43,44]. Therefore, to endure Fe deficiency stress, the capacity to remove excessive ROS is fundamental for plants, where superoxide dismutase (SOD) and peroxidase (POD) activities were of great significance [45]. According to the reports, plants with higher tolerance to Fe deficiency were commonly considered to be equipped with increased SOD and POD activities, and lower ROS accumulation under Fe deficiency stress [46,47]. Nevertheless, it remains to be explored whether *GH3*-modulated FDR is associated with ROS.

While it has been more than a decade since the release of the first genome of pear [48], the knowledge of the pear *GH3* gene family (*PbrGH3s*) is absent, and whether and how *PbrGH3s* are engaged in the FDR of plants have not been explored. Here, the *GH3* gene family from pear (*Pyrus bretschneideri*) was isolated to explore the family members, gene structure, conserved motifs, phylogenetic evolution, *cis*-acting regulatory elements, tissue-specific expression properties, and their expression patterns in the normal (NL) and Fe-deficient chlorotic leaves (CLs) of pear trees. Combined with published RNA-seq data and qRT-PCR analysis, we found that *PbrGH3.5* was strongly induced by Fe deficiency. We then hypothesized that *PbrGH3.5* might participate in the response to Fe deficiency. Subsequently, the function of *PbrGH3.5* was investigated through a reverse genetic approach, and the results showed that the overexpression of *PbrGH3.5* can reduce the Fe deficiency tolerance of tomato plants, which are model plants that share the same Fe absorption system with pear trees and are widely used for Fe deficiency analysis [49,50]. These findings unveiled the crucial mechanisms of plant *GH3*-mediated responses to Fe deficiency, and provided valuable information and a theoretical framework for the functional investigation of pear *GH3* gene members.

## 2. Results

### 2.1. Characterization and Physicochemical Properties of the Pear GH3 Gene Family

A total of 18 candidate pear GH3 proteins (PbrGH3s) were isolated by blasting against the proteome FASTA of the Chinese white pear with *Arabidopsis* AtGH3 and apple MdGH3s as the references, among them, three sequences were duplicated with each other, and another two sequences were overlapped. Therefore, 15 unique members of the GH3 gene family were obtained in pear (Appendix A). These proteins were 133–614 amino acid (aa) long and possessed a molecular weight ranging from 15.26 kDa to 69.58 kDa. The isoelectric points (pI) of the 11 members of PbrGH3s were below 7, while that of the rest of the four members (Pbr006240.1, Pbr021158.1, Pbr034086.1, and Pbr041132.1) were above 7, implying that PbrGH3s are largely acidic proteins. Additionally, 13 of 15 PbrGH3s were predicted to be unstable proteins, as the instability index (II) of them was more than 40, whereas Pbr006240.1 and Pbr021060.1 were identified as stable proteins. Moreover, since the grand average of hydropathy (GRAVY) values of these 15 proteins were all less than 0, they were all considered to be highly hydrophilic. Furthermore, subcellular localization prediction results revealed that the functional location of these 15 PbrGH3 proteins was in the cytoplasm (11), chloroplasts (3), and nucleus (1) (Appendix A), respectively.

### 2.2. Phylogenetic Evolution, Gene Structure, and Motif Analysis of the Pear GH3 Gene Family

To gain a deep understanding of the evolutionary patterns of the PbrGH3s, an maximum-likelihood (ML) phylogenetic tree was constructed among the GH3s from pear, apple, and *Arabidopsis*. The results disclosed that all these proteins were classified into three groups: GH3 I, GH3 II, and GH3 III (Figure 1). Among them, no PbrGH3s and MdGHs were distributed in GH3 III, where only 10 AtGH3 members were included, implying the unique performances of these AtGH3s in some specific physiological processes (Figure 1). In addition, it was found that the distribution pattern of PbrGH3s and MdGHs was parallel. Most GH3 members from pear and apple were involved in GH3 II, and only three PbrGH3s and four MdGH3s were encompassed in GH3 I, reflecting a certain functional similarity between PbrGH3s and MdGH3s (Figure 1 and Figure 2a). Hence, GH3 proteins in pear were renamed by following their evolutionary link to that in apple plants (Appendix A; Figure 1).

Then, we investigated the conserved domain of these PbrGH3s, and the results showed that the domain distribution was similar for all proteins, with a typical GH3 or GH3 superfamily domain span between the N-terminal and C-terminal (Figure 2b). Moreover, it was found that at least 10 motifs were highly conserved in 13 PbrGH3 members by conducting motif analysis (Figure 2c), suggesting their diverse and redundant functions in multiple biological processes. However, unlike the aforementioned proteins that were rich in motifs, only one and three motifs were discovered for PbrGH3.4 and PbrGH3.10B (Figure 2c), respectively, implying their single functions.

### 2.3. Analysis of Cis-Acting Regulatory Elements in the Promoters of the Pear GH3 Gene Family

To investigate the potential functions of *PbrGH3s*, the *cis*-acting regulatory elements (CREs) of the 2000 bp sequences upstream of their CDS (Appendix A) were scanned and the results exhibited that in addition to common shared light-responsive CREs, a variety of hormone-related CREs were identified in the promoters of *PbrGH3s* (Figure 3). Noteworthy, the CREs involved in ABA responses were shared by all *PbrGH3s* and were enriched in them. Interestingly, no ETH-responsive CREs were presented in their promoter regions (Figure 3). Likewise, several stress-involved CREs were detected in the promoters of *PbrGH3s*, and the drought-associated CREs were widely distributed among them (Figure 3). Apart from these findings, some CREs relevant to plant metabolism and development, such as seed-specific regulation and zein metabolism, were also discovered in the promoters of a few *PbrGH3s* (Figure 3). These results indicate that *PbrGH3s* might function in abiotic stress resistance and hormones-mediated progresses.

### 2.4. Expression Analysis and Identification of Fe-Deficiency-Responsive GH3 Gene Family in Pear

To explore the tissue-specific expressions of *PbrGH3s* in pear, the publicly available RNA-seq data of 11 samples from 7 pear tissues was introduced, and the results showed that *PbrGH3.5*, *PbrGH3.6*, and *PbrGH3.12* were expressed across all the tissues surveyed, with *PbrGH3.12* being more highly expressed (Appendix A), suggesting their potential functions in regulating the growth and development of various tissues in pear. Of course, a variety of *PbrGH3s* (*PbrGH3.1*, *PbrGH3.2*, *PbrGH3.3*, *PbrGH3.4A*/*B*, *PbrGH3.10A*/*B*/*C*, and *PbrGH3.14*) were rarely expressed in pear fruits and leaves at different developmental stages; more often, they preferred to express in the ovaries, stems, and shoots of pear plants (Appendix A), indicating their essential roles in the initial occurrence of fruits and leaves.

Consistently, by checking the detail of previously established RNA-seq data on pear normal (NLs) and Fe-deficient chlorotic leaves (CLs), we found similar expression trends for *PbrGH3.1*, *PbrGH3.2*, *PbGH3.3*, *PbrGH3.5*, *PbrGH3.4A*/*B*, *PbrGH3.10A*/*B*/*C*, *PbrGH3.12*, *PbrGH3.13,* and *PbrGH3.14* in leaves (Appendix A). Surprisingly, we noted significant differences in the expressions of *PbrGH3.5*, *PbrGH3.8*, and *PbrGH3.12* in NLs and CLs, with *PbrGH3.5* and *PbrGH3.8* being highly expressed in CL, and the opposite for *PbrGH3.12* (Appendix A). These observations revealed that *PbrGH3.5*, *PbrGH3.8*, and *PbrGH3.12* might participate in Fe deficiency responses (FDR) in pear, and given the most pronounced fold change of *PbrGH3.5* in NLs and CLs (Appendix A), we, therefore, investigated the role of *PbrGH3.5* in FDR.

*PbrGH3.5* is 1806 bp long with an open reading frame (ORF) encoding 601 amino acid residues. Multiple-sequence alignment analysis showed that it had the closest affinity to MdGH3.5 in apple and AtGH3.7 in *Arabidopsis* with similarities of 97.84% and 92.04%, respectively (Appendix A). Consistent with most GH3s in other plants, PbrGH3.5 possesses ligands like adenosine monophosphate (AMP) and 1H-indol-3-Yacetic acid (IAC) in its three-dimensional structure (Appendix A). More importantly, in line with the RNA-seq analysis of NLs and CLs, the qRT-PCR results revealed that compared to the control, the expression of *PbrGH3.5* continued to increase from 1 to 6 h after Fe deficiency treatment, and peaked at 6 h. However, with time prolonging, it rapidly dropped from 6 to 12 h upon Fe deficiency treatment (Appendix A). Moreover, compared to roots, *PbrGH3.5* preferred to express in stems, followed by leaves (Appendix A). These inputs further point out that it may be a key regulator in the FDR of pear.

### 2.5. Functional Analysis of PbrGH3.5 in Fe Deficiency Responses in Tomato Plants

Due to the limitation concerning the pear genetic transformation system, we turned to evaluate the role of *PbrGH3.5* in FDR by overexpressing it in tomato plants (*Solanum lycopersicum*), which share the same manner Fe uptake as pear trees. Subsequently, lines 2, 6, and 7 with higher expression of *PbrGH3.5* (OE2, OE6, and OE7, namely OE plants) were selected for further study (Appendix A). The results displayed that under normal conditions, all plants grew well and maintained green leaves in sufficient chlorophyll content with comparable chlorophyll fluorescence and *Fv*/*Fm* ratio (Figure 4a–d). However, when challenged by Fe deficiency stress, the leaves of both the control (wild-type, WT) and OE plants exhibited noticeable chlorosis, whereas OE plants displayed a more severe degree of chlorosis (Figure 4a). Indeed, while the chlorophyll content and *Fv*/*Fm* ratio in WT and OE leaves were significantly decreased following Fe deficiency treatment, the level was notably lower for OE plants, as well as the bluer chlorophyll fluorescence (Figure 4a–d).

In addition to these observations, we also found that compared to regular conditions, all plants possessed a higher MDA and REL level under Fe starvation stress, and these parameters were notably higher in OE plants than in WT plants (Figure 4e,f). Moreover, by analyzing the ferric chelate reductase (FCR) activity under both conditions, we discovered that the FCR activity was evidently stronger for all plants under Fe deficiency conditions, but it was remarkedly weakened in OE plants, as indicated by the markedly decreased red hue in the FCR assay solution and prominently reduced FCR values (Figure 4g,h). More importantly, pH measurements showed that the pH around the roots of WT and OE plants was fairly constant under control conditions (Appendix A), whereas it clearly dropped after Fe deficiency treatment, and with time prolonging, the rhizosphere pH for OE was markedly higher than that of WT (Figure 4i), indicating that the rhizosphere acidification activity was declined due to *PbrGH3.5* overexpression. Collectively, these results reveal that *PbrGH3.5* is a negative regulator in the FDR of plants.

### 2.6. Dissection of Potential Pathways of PbrGH3.5-Impaired Fe Deficiency Tolerance

To explore the potential working pathways of *PbrGH3.5*, the expressions of the Fe uptake-related genes in tomato plants were examined. No remarkable differences were detected in the expressions of *SlHA4*, *SlFRO1*, *SlIRT1*, and *SlFER* within WT and OE seedlings under Fe-sufficient conditions (Figure 5a–d). However, upon exposure to Fe-deficient treatment, all gene expressions were notably up-regulated, and compared to that of WT, the expressions of these four genes were prominently lower in OE seedlings (Figure 5a–d). These results imply that *PbrGH3.5* overexpression impaired the expressions of Fe-absorption genes in tomato plants, thus contributing to their sensitivity to Fe deficiency.

We then turned to analyze the ROS accumulation in WT and OE seedlings under different conditions, and it was found that the H_2_O_2_ and O_2_^−^ content, two main forms of ROS, did not differ from each other under regular conditions, but after being subjected to Fe deficiency stress, the H_2_O_2_ and O_2_^−^ levels significantly increased, with a dominantly higher amount in OE seedlings (Figure 5e,f). On the contrary, as two vital enzymes responsible for ROS scavenging, the SOD and POD activities were dominantly stronger in both WT and OE plants under Fe-deficient conditions, and the activities were strikingly higher for WT plants, whereas they were comparable in WT and OE lines under Fe-sufficient conditions (Figure 5g,h). These outcomes hint that the overexpression of *PbrGH3.5* in tomato plants weakens their ability to scavenge ROS.

To investigate the relationship between the *PbrGH3.5*-diminished FDR with IAA, we subsequently determined the IAA content in WT and OE seedlings under Fe deficiency stress. The results showed that when challenged by Fe deficiency, the free IAA content in WT plants was markedly higher than that in OE plants (Appendix A). However, when comparing the conjugated IAA, we observed that while the content of IAA-leucine (Leu) did not differ between OE and WT plants, the content of IAA-phenylalanine (Phe) was significantly decreased, and IAA-glutamate (Glu) was pronouncedly increased in OE seedlings than that in WT (Appendix A). And compared with IAA-Glu, the content of IAA-Phe was greatly lower (Appendix A). These findings illustrate that the decline in IAA among *PbrGH3.5*-overexpressed tomato plants may account for their Fe deficiency sensitivity.

Taken together, *PbrGH3.5*-suppressed Fe deficiency tolerance was associated with diminished Fe acquirement, compromised ROS scavenging, and declined free IAA accumulation.

### 2.7. Prediction of the Interaction and Regulatory Networks for PbrGH3.5

To explore the proteins that interact with PbrGH3.5 or regulate the expression of *PbrGH3.5*, the potential interactors and transcription factors (TFs) were predicted through the STRING and PlantTFDB online websites with MdGH3.5 as a reference, followed by homology blast in the proteome of Chinese white pear. A total of nine tentative interactors were identified, and based on their annotation, SNF1-related protein kinases 2 (SnRK2) and *Arabidopsis* response regulators (ARRs) were enriched (Figure 6a), demonstrating that PbrGH3.5 might integrate ABA and CTK signaling to regulate relevant biological processes, such as FDR. In the predicted regulation network, up to 20 of the TFs were pooled, including DNA binding with one finger (Dof), v-myb avian myeloblastosis viral oncogene homolog (MYB), basic (region) leucine zippers (bZIP), and MADS. Among them, Dofs were the most overrepresented (Figure 6b), and supporting this, many Dof-binding sites were found in the promoter of *PbrGH3.5* (Appendix A), implicating the pivotal role of Dof TFs in FDR.

## 3. Discussion

### 3.1. Pear GH3 Gene Family Consists of 15 Members

In this study, we initially identified 15 potential candidates for the Chinese white pear *GH3* gene family (Appendix A; Figure 1 and Figure 2), which are roughly equal in number to the *GH3* members in apple, maize, and tea [29,30,34], but less than that in wheat and potato [32,33]. However, similar to the physical properties of MdGH3s [29], most PbrGH3s encoded proteins with a predicted length of 570–620 aa, and molecular weight of 63.62–69.58 kD, except for PbrGH3.10B, PbrGH3.10C, and PbrGH3.14 (Appendix A). In addition, according to the pI, II, and GRAVY results (Appendix A), we concluded that most PbrGH3s belong to active proteins that are acidic and hydrophilic (Appendix A), which is in line with the chemical properties of TaGH3s and GhGHs [31,32].

Parallel to the structure of GH3s in other plant species [30,34,51], these 15 pear GH3 proteins investigated all have a typical GH3 or GH3 superfamily domain throughout the center (Figure 2b), and most of them share a variety of motifs (Figure 2c), which are imperative for their natural functions. These results suggest that although the members of the *GH3* gene family vary among plant species, its structure remains relatively conserved, implying the presence of functional similarities in the *GH3s* between species. Moreover, based on the analysis of the ML tree, GH3 gene family from pear, apple, and *Arabidopsis* could be divided into three groups, and just like the evolution pattern of MdGH3s [29], PbrGH3s were also presented in Groups I and II, and were widely distributed in the latter (Figure 1 and 2a). More importantly, PbrGH3s and MdGH3s were all absent in Group III, a unique type shared by AtGH3s [24], CmGH3s [52], GmGH3s [31], and other GH3s. These findings address that the GH3 members in apple and pear may have originated from a single ancestral gene and also indicate a differentiation among *GH3s* in species.

### 3.2. PbrGH3s May Participate in ABA-Mediated Abiotic Stresses

Here, we analyzed the CREs in the promoters of *PbrGH3s* and found that various CREs associated with responses to light and hormones existed in their upstream regions. Notably, apart from IAA-affiliated CREs, a significant number of ABA-responsive CREs were displayed in the promoters of all *PbrGH3s*, and most *PbrGH3s* contained a variety of drought-related CREs (Figure 3). CREs are the direct binding site of appropriate transcription factors [53,54], reflecting the physiological processes in which the target genes may be involved and offering the possibility to predict their hidden functions [55,56]. Therefore, in combination with the established reports that ABA is a stress-responsive hormone widely involved in drought stresses [57,58] and that both apple *MdGH3s* and grape *VvGH3.9* gene were responsible for their drought adaptation [40,59], *PbrGH3s* could be primarily embroiled in the ABA-regulated pathways, such as drought responses.

### 3.3. PbrGH3.5 Negatively Modulates Fe Deficiency Responses in Tomato Plants

In this work, we studied the tissue-specific expression pattern of *PbrGH3s*, and the results showed that unlike the expression patterns of other *GH3s* [33,34], most *PbrGH3s* were rarely expressed in fruits and leaves, except for *PbrGH3.5*, *PbrGH3.10C*, *PbrGH3.12*, and *PbrGH3.13* (Appendix A). Instead, they favored expression within the ovaries and buds (Appendix A), which is similar to that of maize [30]. Likewise, by surveying their expressions in pear normal (NLs) and Fe-deficiency-caused chlorotic leaves (CLs), it was discovered that consistent with the tissue-specific expression patterns, only a few *PbrGH3s* were detected in both NLs and CLs. Among them, *PbrGH3.5* showed the highest fold change in the comparison of CL vs. NL (Appendix A).

Tissue-specific expression is the critical trait of genes, reflecting the functional locations of the corresponding genes, and thus indicating the relevant biological progress they involved [60]. For example, rice *Auxin Response Factor12* (*OsARF12*) with root tip expression predisposition was responsible for the root elongation and iron acquisition [61], and bamboo *ROOTHAIRLESS4* (*PeRHL4*) with root expression preference was in charge of phosphorus starvation and drought responses [62]. Therefore, we concluded that while the function of *GH3s* might vary among species, there are shared functions they contribute to, where the majority of *PbrGH3s* may not account for the development of fruits and leaves. Meanwhile, since IAA and its early responsive genes play vital roles in pear FDR [63], and the function of *GH3* genes in response to abiotic stresses had been generally reported [39,40,41], *PbrGH3s* might be involved in responses to Fe deficiency, especially for *PbrGH3.5*.

### 3.4. PbrGH3.5-Weakened Fe Deficiency Tolerance in Tomato Plants Is Related to ROS Accumulation and IAA Conjugation

In line with the RNA-seq data, *PbrGH3.5* was significantly induced under Fe starvation stress at the early stage but was notably inhibited hereafter (Appendix A), illustrating that *PbrGH3.5* did respond to Fe deficiency. Indeed, by overexpressing this gene in tomato plants, we discovered that the leaves of the transgenic seedlings were yellower and damaged more seriously than WT under Fe-limited conditions, accompanied by notably decreased chlorophyll content, and a lower chlorophyll fluorescence and *Fv*/*Fm* ratio (Figure 4a–f). More importantly, both the rhizosphere acidification and FCR abilities, the determining factors in FDR [64,65,66], were significantly weakened in OE seedlings (Figure 4g–i). This might be due to the prominently repressed expression of *SlHA4*, *SlFRO1*, *SlIRT1*, and *SlFER* (Figure 5a–d), which were hub genes responsible for pH reduction, ferric reduction, and Fe absorption in tomato [50,63]. These results unveil that *PbrGH3.5*-overexpressed seedlings were more sensitive to Fe deficiency stress, which might be caused by the reduced expressions of Fe-uptake-related genes.

Following further determination of physiological indicators, we ascertained that Fe deficiency induced the accumulation of the ROS in tomato plants, and the ROS contents, including H_2_O_2_ and O_2_^−^, in Fe deficiency-sensitive transgenic tomato plants were markedly higher in OE plants than WT plants under Fe-deficient conditions (Figure 5e,f). In addition, we uncovered that the SOD and POD activities were all strengthened by Fe deficiency, whereas they were all less active in OE seedlings (Figure 5g,h). However, while ROS are signaling molecules with both beneficial and harmful effects in plants [42,67], they are overproduced under Fe deficiency stress, and the excessive ROS are more inclined to be the toxic byproducts [43,68,69], triggering oxidative damage and even programmed cell death [44]. SOD and POD activities are two key enzymes to scavenge over-accumulated ROS and play positive roles in resisting Fe deficiency stress [45]. Therefore, the impaired ROS scavenging in *PbrGH3.5*-overexpressed tomato plants might account for their greater hypersensitivity to Fe deficiency.

Beyond these findings, we also noticed that the free IAA contents in OE plants were markedly decreased compared to WT under Fe-starved conditions, and the level of IAA-Glu, a main form of conjugated IAA [27], was significantly higher in OE seedlings after Fe deficiency treatment (Appendix A). These results illustrate that *PbrGH3.5*, a typical Group II member with AMP and IAC ligands (Appendix A), could facilitate the conjugation of IAA to amino acids under Fe-limited stress, which is consistent with the fact that *GH3* members in Group II have IAA-amido synthetases, and could affect the homeostasis of free IAA and conjugated IAA [24,70]. Because of the positive functions of free IAA in enhancing the tolerance of plants to Fe deficiency [20,23], the increased IAA conjugation might be another influential factor in *PbrGH3.5*-diminished Fe deficiency tolerance in tomato plants. Importantly, given the spatiotemporal expression profiles of IAA synthesis genes [71], it is sound that it is not the decrease in total free IAA content, but the reduction in spatial IAA at specific positions that might weaken the FDR of the OE seedlings, which needs further study.

### 3.5. Interactor PbrSnRKs and Transcription Factor PbrDofs Might Determine the Function of PbrGH3.5

Here, we performed a prediction of the interaction network of PbrGH3.5, and found that PbrGH3.5 might interact with PbrSnRKs (Figure 6a). SnRK is a key part of ABA signaling and functions in enhancing the tolerance of plants to Fe deficiency by promoting Fe absorption [72]. Therefore, PbrGH3.5 might manage FDR by integrating ABA and IAA signals through the interaction with SnRKs in plants. Additionally, given that SnRKs could act as phosphorylation to enhance stability and inhibit/promote the activity of co-factors through the interaction [57,73]; thereby, SnRKs-developed phosphorylation might be critical for the function of PbrGH3.5, and on account of the reciprocal effect of the interaction, PbrGH3.5 might, in turn, participate in protecting SnRK from degradation.

Likewise, we carried out a regulatory network analysis of *PbrGH3.5* and noted that *PbrGH3.5* might be the downstream target gene of PbrDof TFs (Figure 6b), which was further supported by the existence of quite a lot of potential binding sites in its upstream regions (Appendix A). Dof is a classic and specific TF family in plants [74], and is addressed to regulate the growth and development, and responses to abiotic stresses in plants, such as root development [75], leaf senescence [76], drought stress [77], and salinity stress [78]. Although the study of the relationship between Dof and FDR is rarely reported, some works have found that the ability of plants to resist Fe deficiency is connected with root development to a certain extent [23,79]. Consequently, the PbrDof-*PbrGH3.5* transcriptional loop may function in pear FDR.

## 4. Materials and Methods

### 4.1. Identification and Bioinformatics Analysis of PbrGH3 Genes

The sequences of 20 reported *Arabidopsis* GH3 proteins (AtGH3s) [24] and 16 reported apple GH3 proteins (MdGH3s) [29] were separately utilized as the query to search the target *GH3* gene family from the proteome FASTA of Chinese white pear (http://peargenome.njau.edu.cn/, accessed on 28 November 2024) [48] using the TBtools v1.119 [80], and the screening criteria were set as E-value ≤ 1 × 10^−5^. The results of the two search methods were then combined, and the tentative genes were retained by removing the redundant members. Next, to vilify the presence of the conserved domain PL03321 (GH3), the amino acid sequences of the putative proteins were subjected to the Batch CD-Search tool (https://www.ncbi.nlm.nih.gov/Structure/bwrpsb/bwrpsb.cgi, accessed on 28 November 2024), and to examine their conserved motifs, the MEME tool (https://meme-suite.org/meme/tools/meme, accessed on 28 November 2024) was applied. The sequences of AtGH3s and MdGH3s were accessed from the online website Phytozome (https://phytozome.jgi.doe.gov/pz/portal.html, accessed on 28 November 2024). TBtools v1.119 was used to graphically visualize the conserved domains and motifs of GH3 proteins from the pear. The protein IDs of GH3s used here are listed in Appendix A.

Alongside this, the physicochemical features of the resulting PbrGH3s were calculated using the online tool Protparam (https://web.expasy.org/protparam/, accessed on 28 November 2024), including the number of amino acids (Length), molecular weight (MW), theoretical isoelectric points (pI), instability index (II), aliphatic index (AI), and the grand average of hydropathicity (GRAVY). Moreover, the protein sublocation was predicted using the Cell-PLoc 2.0 web server (http://www.csbio.sjtu.edu.cn/bioinf/Cell-PLoc-2/, accessed on 28 November 2024), and the tridimensional structure of the PbrGH3.5 was constructed using the online program Swiss model (http://swissmodel.expasy.org/, accessed on 28 November 2024).

### 4.2. Phylogenetic and Expression Profile Analysis of PbrGH3 Genes

To explore the evolutionary relationships of the PbrGH3s, a maximum-likelihood (ML) phylogenetic tree was constructed among PbrGH3s, MdGH3s, and AtGH3s using the ClustalW program in MEGA v7.0 software (https://www.megasoftware.net, accessed on 28 November 2024) under the LG model, and the resulting tree was visualized and beautified using the EvolView tool (https://www.evolgenius.info/evolview-v2/, accessed on 28 November 2024). Based on their distribution profiles, the multiple-sequence alignment was performed among PbrGH3.5, MdGH3.5, and AtGH3.7 using the DNAMAN 9.0 software (https://www.lynnon.com/dnaman.html, accessed on 28 November 2024).

To gain an insight into the tissue-specific expression profiles of *PbrGH3s* in pear, the retrieved RNA-seq data on the online website PearMODB (https://pearomics.njau.edu.cn/, accessed on 28 November 2024) [81] were exploited. The encompassed tissues were the leaves and fruits of 4, 6, 8, and 10 weeks after flower blooming, petals, calyxes, ovaries, stems, and buds, for a total of 11 samples from 7 samples. All tissues were harvested from different periods of the same pear tree (Dangshansuli, a species of the white pear that was widely cultivated locally) in an orchard located in the Old Yellow River Valley region in Dangshan County, Anhui Province (Suzhou, China) [82]. In addition, the expressions of *PbrGH3s* in the normal (NLs) and Fe-deficient chlorotic leaves (CLs) of pear trees were analyzed by referring to a previous work completed by our lab [63]. As described, the NLs and CLs were, respectively, collected from the 55-year-old normal pear trees and Fe-deficiency-stressed pear trees (Dangshansuli). For the latter, the available Fe was limited due to the high pH value of the local soil [63]. The expression profiles were presented in a heatmap using GraphPad Prism 9.5 (https://www.graphpad.com/scientific-software/prism/www.graphpad.com/scientific-software/prism/, accessed on 28 November 2024).

### 4.3. Promoter Analysis and Regulatory Prediction of PbrGH3 Genes

To survey the *cis*-acting regulatory elements (CREs) in the promoters of *PbrGH3s*, a length of the 2000 bp region upstream of the start codon of the corresponding genes was searched from the genomic DNA sequences of the Chinese white pear and scanned on the online PlantCARE database (https://bioinformatics.psb.ugent.be/webtools/plantcare/html/, accessed on 28 November 2024); the resulting CREs were summed and divided into three categories: hormone responsiveness, biotic and abiotic stress, and metabolism and development, and finally presented in the heatmap. Given the lack of relevant data information of pears on the corresponding websites, the orthologue of PbrGH3.5 in apple, MdGH3.5, was instead used to explore the potential interactors of PbrGH3.5 and the interaction partners of MdGH3.5 were, respectively, analyzed using the STRING tool (https://cn.string-db.org/, accessed on 28 November 2024). To investigate the possible regulatory TFs of *PbrGH3.5*, its promoter was subjected to Plant Transcription Factor Database 5.0 (http://planttfdb.cbi.pku.edu.cn/, accessed on 28 November 2024), and the TFs were inquired against the apple (*M. domestica*) databases. The pear protein IDs were finally accessed from the proteome of Chinese white pear by performing a blasting program in BioEdit v7.5.0.3 (https://thalljiscience.github.io/, accessed on 28 November 2024) against the corresponding apple orthologues. Cytoscape v3.5.1 (https://cytoscape.org/download.html, accessed on 28 November 2024) was used to construct the interaction and regulatory networks.

### 4.4. Plant Materials and Fe Deficiency Treatment

Wild-type tomato (*Solanum lycopersicum*) *Micro-Tom* was used in this study. Tomato seeds were surface-sterilized with 50% sodium hypochlorite solution for approximately 10 min. Then, the seeds were rinsed with distilled water for 30 s and repeated three-to-four times. Hereafter, the aseptic seeds were air-dried on the sterile filter paper and subsequently germinated on an MS base salt (Coolaber, Beijing, China) with an additional 3% sucrose (*w*/*v*) and 0.3% phytol (*w*/*v*) adjusted to pH 5.8 in darkness for two days. Finally, the seeds were allowed to grow in a growth chamber at 25 °C for two weeks before genetic transformation under long-day conditions: 16 h light/8 h dark with 80% relative humidity and 12,000 µmol∙m^−2^∙s^−1^ light intensity. For normal soil growth, the seeds stored at 4 °C were sowed into pots with culture substance (nutrient soil: vermiculite = 1:1, *v*/*v*), and cultured under environment-controlled conditions.

To evaluate the Fe deficiency tolerance of tomato plants, one-month-old soil-cultured tomato seedlings were removed from the pots and washed with running water several times to clean the roots. Next, the seedlings were fixed in black hydroponic boxes containing a Hoagland nutrient solution (NS10205, Coolaber, Beijing, China) to allow them to adapt to the hydroponic conditions. Three days later, the seedlings were separately subjected to Fe-sufficient (Hoagland nutrient solution, +Fe) and Fe-deficient (NS1010, Hoagland nutrient solution without Fe, −Fe) treatments. The phenotype was monitored in the following days [49], and the chlorophyll-related indicators were detected until phenotypic differences. Hereafter, the roots and the top two leaf samples were subsequently collected and used for the determination of physiological indexes and analysis of the expressions of genes involved in Fe acquisition [50], as well as the IAA content.

To examine the expression pattern of *PbrGH3.5* under Fe-deficient stress, ‘Duli’ pear seeds (*P. betulifolia*) were applied and were firstly sand-stratified at 4 °C for a month, and then sowed in pots with culture substance for germination and growth in a greenhouse. One month later, the pear seedlings were removed from the pots and washed with distilled water. Afterward, the cleaned seedlings with uniform growth were subjected to Fe-deficient treatment as carried out for the tomato seedlings above. Eventually, the 2nd and 3rd leaf samples were collected at 0, 1, 3, 6, 9, and 12 h after the treatment, and stored at −80 °C for quantitative real-time PCR (qRT-PCR) analysis. Each point contains three independent repeats. For the expression analysis of *PbrGH3.5* in pear tissues, the leaves, stems, and roots of one-month-old ‘Duli’ pear seedlings grown on soil were sampled and subjected to qRT-PCR detection.

### 4.5. Plasmid Construction and Plant Transformation

The coding sequence (CDS) of *PbrGH3.5* was cloned from the leaves of ‘Dangshansuli’ and inserted into the p1300-35S-GFP-BS2 vector using the primer pairs of p1300_PbrGH3.5-F/R (Appendix A). The recombinant vector, p1300-*PbrGH3.5*, was then transformed into the *Agrobacterium tumefaciens* strain GV3101-pSoup via the thermal stimulation method. *Agrobacterium*-mediated leaf disk transformation was performed to obtain transgenic tomato plants overexpressing *PbrGH3.5,* as previously described [83]. Briefly, the cotyledons of two-week-old tomato seedlings were cut into small pieces using a sterile scalpel and soaked in a suspension containing the strains with the recombinant vector at 25 °C for 10 min, and the infiltrated pieces were air-dried on the sterile filter paper. After that, they were placed on the co-culture medium (MS medium + 1 mg∙L^−1^ 6-Benzylaminopurine + 0.1 mg∙L^−1^ IAA) in darkness for 2 d, and transferred to a selecting medium (MS medium + 1 mg∙L^−1^ zeatin + 0.1 mg∙L^−1^ IAA + 25 mg∙L^−1^ hygromycin + 400 mg∙L^−1^ timentin) until the positive shoots appeared. Since then, the transgenic shoots were validated on the DNA and transcription levels and subsequently subjected to MS medium supplied with 2 mg∙L^−1^ indole-3-butyric acid for rooting. Three independent lines were chosen for the phenotypic analysis.

### 4.6. Physiological Indexes Analysis

The total chlorophyll content was measured using the acetone extraction method following previous work with some modifications [84]. In short, 0.1 g sample of tomato leaves was cut into pieces and decolored in an extraction solution (50% (*v*/*v*) acetone + 40% (*v*/*v*) absolute alcohol + 10% (*v*/*v*) distilled water) overnight, and the absorbance of the extraction solution was detected at 663 nm and 645 nm using with a spectrophotometer (V-T3/V-T3C; Yipu, Shanghai, China), respectively, and subsequently recorded as A1 and A2. Finally, the total chlorophyll content was calculated using the following formula: 8.04A1 + 20.29A2.

The relative electrolyte leakage (REL) of the leaves was determined using a conductivity meter (a-AB33ECZH, OHAUS, Parsippany, NJ, USA) according to the method described in a previous study [85]. In brief, eight equal-sized leaf pieces were immersed in a 15 mL plastic tube with 10 mL of the distilled water and rotated in a rotation meter at the speed of 200 rpm for 2 h, and the conductivity was recorded as S2, while the initial conductivity of the water was determined as S1. Then, the tube was dipped in boiled water for 30 min, and cooled to room temperature under running water, and the final conductivity was measured as S3. The REL was counted as follows: (S2 − S1)/(S3 − S1) × 100%.

The chlorophyll fluorescence of the *Fv*/*Fm* ratio was imaged using an IMAGING-PAM instrument (WALZ Co., Nuremberg, Germany), and the *Fv*/*Fm* ratio was analyzed using ImagingWin software (https://www.walz.com/products/chl_p700/imaging-pam_ms/downloads.html, accessed on 28 November 2024) [86]. The contents of malondialdehyde (MDA), peroxide (H_2_O_2_), and superoxide radical (O_2_^−^), and the activities of superoxide dismutase (SOD, EC 1.15.1.1), peroxidase (POD, EC 1.11.1.7) of tomato leaves were measured according to the instructions of commercially available kits (Solarbio, Beijing, China) [87]. The product catalogs are BC0025, BC3595, BC1295, BC5165, and BC0095, respectively. The ferric chelate reductase (FCR) activity was analyzed using the assay buffer solution (0.1 mM Fe(III)-EDTA and 0.3 mM Ferrozine), as previously described [63]. The biomass of both the primary and lateral roots of the tested tomato seedlings was weighted (W, g) and immersed into the assay solution in darkness for 2 h, and the absorbance of it was read (A) in a spectrophotometer at 562 nm. The FCR activity was quantified using the following equation: 700 A/57.2 W. The pH value of the nutrient solution around the roots was detected using a portable pH meter (ST3100, Ohaus, NJ, USA) [64], and the IAA level was calculated by conducting UPLC-ESI-MS/MS assay of the previously published study [88]. All tests were performed with at least three repetitions.

### 4.7. qRT-PCR Analysis

The total RNA from roots, stems, and leaves of ‘Duli’ pear seedling, the genomic DNA (gDNA), and total RNA from tomato leaves were extracted using corresponding commercial kits (DP320 for gDNA and DP441 for RNA, TIANGEN, Beijing, China). The complementary DNAs (cDNA) used for qRT-PCR were generated using the kit with a genomic DNA (gDNA) remover (R323, Vazyme, Nanjing, China). The qRT-PCR was performed in the reaction buffer containing SYBR Green Master Mix (Vazyme, Nanjing, China) and gene-specific prime pairs on an ABI StepOne qRT-PCR detection system (Applied Biosystems, Foster City, CA, USA). The relative gene expression was calculated based on the 2^−ΔΔCt^ method [89], with pear actin (*PbrActin*) and tomato actin (*SlActin7*) serving as the internal controls. The primers used in this study are listed in Appendix A.

### 4.8. Statistical Analysis

All the experiments were carried out in three independent biological replicates. GraphPad Prism 9.5 software and Microsoft Excel software 2017 were applied to construct the heatmaps and column charts. Data were shown as the mean ± standard deviation (SD). The SPSS v 26.0 software (SPSS Inc., Chicago, IL, USA, https://www.ibm.com/cn-zh/spss, accessed on 28 November 2024) was applied to analyze the statistical differences using a one-way analysis of variance (ANOVA) with Duncan’s multiple-range test or Student’s *t*-test, and the significant differences were indicated by different lowercases at the level of *p* < 0.05.

## 5. Conclusions

Collectively, a total of 15 *GH3* gene members in Chinese white pear (PbrGH3s) were examined and could be grouped into two subclusters. Most of these members are highly conserved in gene structure and protein motifs, and showed tissue-specific expression preferences. *PbrGH3s* might function in physiological processes associated with hormone signaling and abiotic stress, and several *PbrGH3s* may participate in Fe deficiency responses. Among them, *PbrGH3.5*, a potential interactor of PbrSnRKs and putative target downstream of PbrDofs, was characterized to negatively regulate the Fe deficiency responses of tomato plants in ROS- and IAA-dependent manners. Overall, our results established the foundation for further comprehensive investigations of *PbrGH3s*.

## Figures and Tables

**Figure 1 ijms-25-12980-f001:**
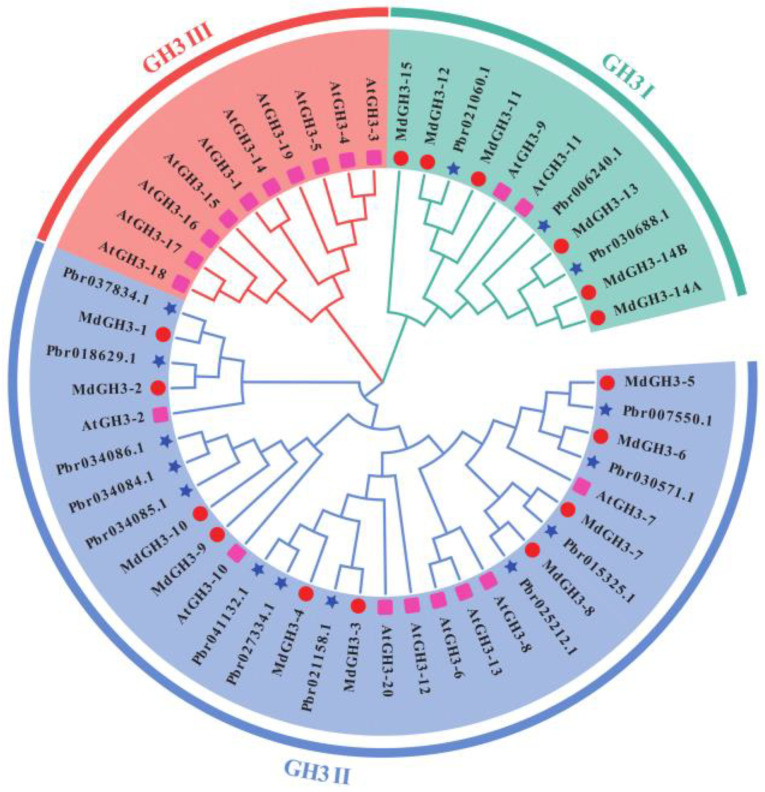
Phylogenetic tree showing the relationships between GH3 gene families from pear, apple, and *Arabidopsis*. The alignments were conducted with ClustalW, and the phylogenetic tree was generated using MEGA7.0 with 1000 bootstrap replications; the maximum-likelihood (ML) method under the LG model was introduced. The resulting tree was visualized and embellished using the online tool (https://www.evolgenius.info/evolview-v2/, accessed on 28 November 2024). Different groups were distinguished with different colors, and the GH3 proteins from pear (*Pyrus bretschneideri*, Pbr), *Arabidopsis* (*Arabidopsis thaliana*, At), and apple (*Malus domestica*, Md) were decorated with blue stars, red circles, and pink squares, respectively.

**Figure 2 ijms-25-12980-f002:**
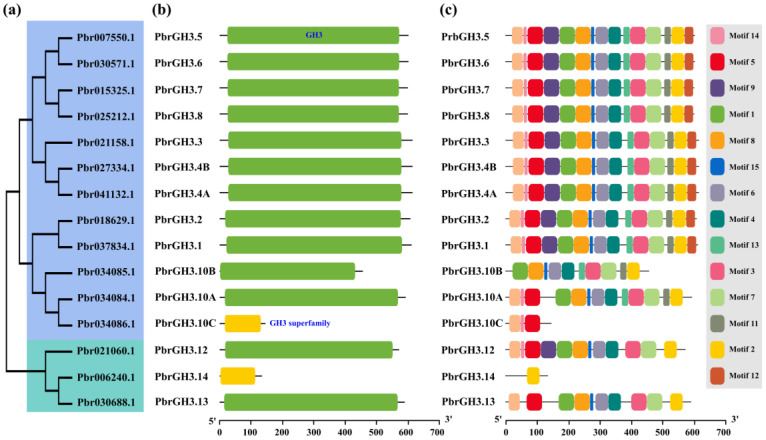
Phylogenetic tree, conserved domain, and featured motifs of pear GH3 proteins (PbrGH3s). (**a**) The maximum-likelihood (ML) evolutionary tree of PbrGH3s. Different groups are marked with different color backgrounds. (**b**) The structure of the conserved domains and (**c**) the distribution of the featured motifs of PbrGH3s. The GH3 and GH3 superfamily domains were displayed in green and yellow boxes, respectively. Motifs 1 to 15 are presented with different color boxes. The numbers below mean the length of the proteins.

**Figure 3 ijms-25-12980-f003:**
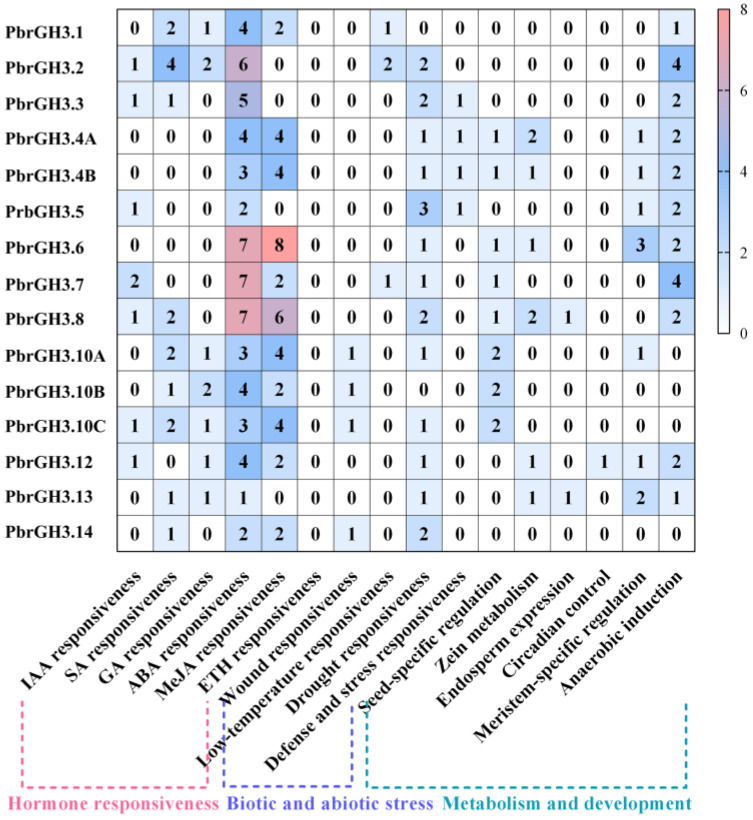
*Cis*-acting regulatory elements (CREs) in the promoters of the *GH3* genes in pear (*PbrGH3s*). The 2000 bp upstream regions of the coding sequences (CDS) of the corresponding *PbrGH3s* were analyzed using the online tool PlantCARE (https://bioinformatics.psb.ugent.be/webtools/plantcare/html/, accessed on 28 November 2024). The numbers in the boxes state the sum of various CREs involved in the same stimuli and are presented in the heatmap. The white-to-orange gradient indicates a gradual increase in number.

**Figure 4 ijms-25-12980-f004:**
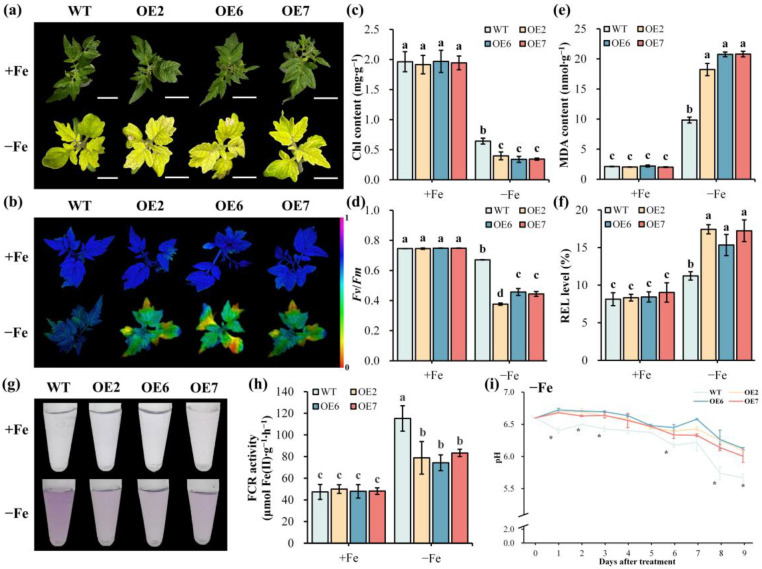
*PbrGH3.5*-overexpressed tomato plants are more sensitive to Fe deficiency stress. (**a**) The phenotype of wild-type (WT) and *PbrGH3.5*-overexpressed transgenic tomato seedlings (OE1, OE6, and OE7, namely OE) under Fe-sufficient (+Fe) and Fe-deficient (−Fe) conditions for approximately 14 days. (**b**) The total chlorophyll content in the leaves of WT and OE seedlings after +Fe and −Fe treatments. (**c**) The chlorophyll fluorescence, (**d**) maximum photochemical efficiency of photosystem II (*Fv*/*Fm*) ratio, (**e**) MDA content, and (**f**) relative electrolyte leakage (REL) of WT and OE leaves after +Fe and −Fe treatments. (**g**) Visualization of root FCR activities in WT and OE lines under +Fe and −Fe conditions. (**h**) The FCR activities of WT and OE roots after +Fe and −Fe treatments. (**i**) The rhizosphere pH over time under −Fe conditions. The data are shown as the mean ± standard deviation (SD) of three biological replicates (n = 3), and the statistical differences are indicated by different lowercases (a–c) or asterisks at *p* < 0.05 (one-way ANOVA with Duncan’s multiple range test for charts (**b**,**d**–**f**,**h**), and Student’s *t*-test for chart (**i**)).

**Figure 5 ijms-25-12980-f005:**
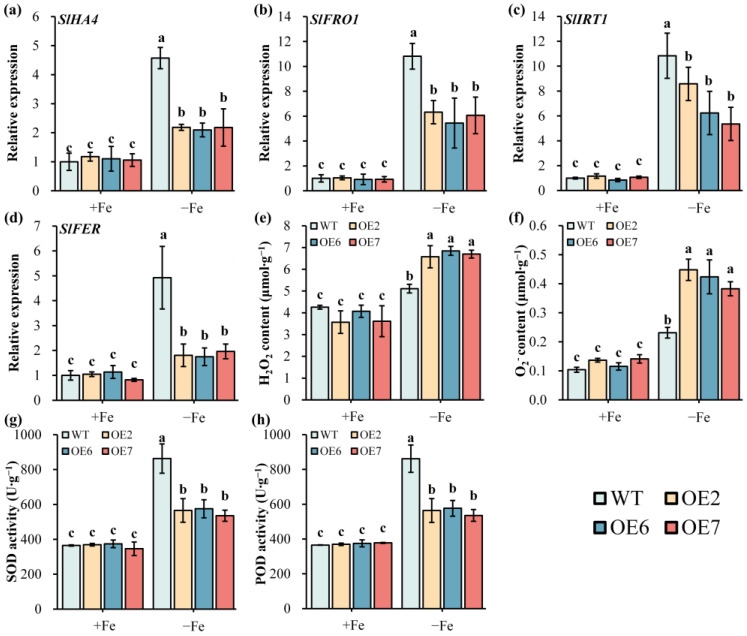
Ectopic overexpression of pear *PbrGH3.5* in tomato plants inhibits their capacities in Fe acquisition and ROS scavenging under Fe deficiency stress. (**a**–**d**) The expressions of *SlHA4*, *SlFRO1*, *SlIRT1*, and *SlFER* in the roots of wild-type (WT) and *PbrGH3.5*-overexpressed transgenic tomato seedlings (OE1, OE6, and OE7) seedlings after Fe-sufficient (+Fe) and Fe-deficient (−Fe) treatments. The expression of the corresponding gene in WT roots under +Fe conditions was used as the control and was set to ‘1’. (**e**) H_2_O_2_ and (**f**) O_2_^−^ content in WT and OE leaves after +Fe and −Fe treatments. (**g**) The SOD and (**h**) POD activities in the leaves of WT and OE seedlings at different conditions. The data are shown as the mean ± standard deviation (SD) of three biologically independent samples (n = 3), and the statistical differences are indicated by different lowercases (a–c) at *p* < 0.05 (one-way ANOVA with Duncan’s multiple range test).

**Figure 6 ijms-25-12980-f006:**
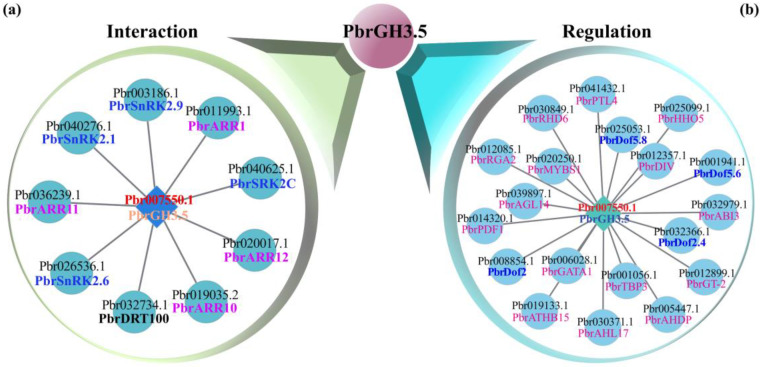
Prediction of the interaction (**a**) and upstream regulation (**b**) network of pear *PbrGH3.5*. The orthologue MdGH3.5 protein in apple (*Malus domestica*) was used as the inquiry to identify the potential interactor of pear PbrGH3.5 protein using the STRING tool (https://cn.string-db.org/, accessed on 28 November 2024), and the possible upstream regulatory transcription factors (TFs) of *PbrGH3.5* was analyzed by subjecting the corresponding promoter to the Plant Transcription Factor Database 5.0 (http://planttfdb.cbi.pku.edu.cn/, accessed on 28 November 2024) with the apple databases as the reference. The pear protein IDs were eventually obtained from the Chinese white pear proteome by performing blast analysis in BioEdit v7.5.0.3 (https://thalljiscience.github.io/, accessed on 28 November 2024) against the corresponding apple orthologues. The related networks were constructed using Cytoscape v3.5.1 (https://cytoscape.org/download.html, accessed on 28 November 2024).

## Data Availability

The original contributions presented in this study are included in the article/Appendix A. Further inquiries can be directed to the corresponding author(s).

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
