# Peer review of "GH3 Gene Family Identification in Chinese White Pear (Pyrus bretschneideri) and the Functional Analysis of PbrGH3.5 in Fe Deficiency Responses in Tomato"

_ijms, 2024, doi:10.3390/ijms252312980_

Round 1
Reviewer 1 Report
Comments and Suggestions for Authors
The manuscript has the important idea but was exposed in a way that is not scientific, or didactic. The essential details about experiment of Fe deficiency are missing, together with those about the organ collection, plant age… They must be included in the beginning of M&M. The use of other species, such as tomato, Arabidopsis or apples, must be explained explicitly. The text is too long, with a lot of information and maybe you can think how to pass those information in two papers, or similar. Or to show less figures/tables and the text, but to preserve the essence.
Keywords cannot repeat the words form the title
The introduction must introduce the general functional traits researched here.
M&M must be organized, statistics must be clear and clean.
Results must be direct. Always indicate where we have to look, in the beginning of the paragraph, and please, describe all details in captions.
Discussion must start with YOUR NOVELTY.
Various details are included in the pdf of the manuscript.
-----------------------------------------------------------------------------------------------------

Author Response
Responses to the reviewer 1
Dear editors and peer reviewers,
On behalf of my co-authors, we would like to thank you for giving us the opportunity to revise our manuscript. We appreciate the editor and the reviewers for the constructive comments on our manuscript entitled “Genome-wide analysis of the GH3 gene family in white pear (Pyrus bretschneideri) reveals the negative function of PbrGH3.5 in Fe deficiency response” (ijms-3311849). Those comments are valuable and helpful for reversing our paper, as well as important to improve our research work.
All the questions proposed by the reviews have been considered carefully and the manuscript has been revised accordingly. The reversion portion is marked in red on the paper. Our “Response to the reviewers’ Comments” were provided in blue fonts in the following text.
Reviewer 1,
Comment 1: The manuscript has an important idea but was exposed in a way that is not scientific, or didactic. The essential details about the experiment on Fe deficiency are missing, together with those about organ collection, plant age, etc. They must be included at the beginning of M&M.
Responses: Thank you for your constructive comments. According to your suggestions, we have supplemented the relevant details in the main text. For example, the details about the experiment of Fe deficiency on ‘Duli’ pear were further improved in Lines 313−327, the procedure to conduct Agrobacterium-mediated leaf disk transformation of tomatoes was added in Lines 337−346, and the methods to detect the physiological indicators were described in Lines 354−399, as well as the details of the collection of test samples in Lines 227−246, and Lines 298−302, and the growth conditions of tomato and pear seedling (Caption 2.4. Plant materials, and Fe deficiency treatment). Moreover, based on your comments given in the original PDF file, the details of the software and apparatus used in this work were complemented in Lines 225, 368, 396, 417, etc. Please find the details in the revised manuscript.
Comment 2: The use of other species, such as tomato, Arabidopsis, or apples, must be explained explicitly.
Responses: Thank you for the kind reminder. The reasons why tomato plants were introduced to investigate the role of PbrGH3.5 in Fe deficiency responses and why apple MdGH3.5 was applied as the target to identify the potential interactor of PbrGH3.5 were presented in Lines 257−261, and Lines 600−609, respectively. The related contents were stated as follows:
1, Given the lack of relevant data information on pears on the corresponding websites, the ortholog of PbrGH3.5 in apple, MdGH3.5, in apple, was instead used to explore the potential interactors of PbrGH3.5.
2, Due to the limitation concerning the pear genetic transformation system, we turned to evaluate the role of PbrGH3.5 in FDR by overexpressing it in tomato plants (S. lycopersicum), which share the same manner to uptake Fe with pear trees.
Please find the details in the revised manuscript.
Comment 3: The text is too long, with a lot of information and maybe you can think about how to pass the information in two papers, or similar. Or to show fewer figures/tables and the text, but to preserve the essence.
Responses: Thank you for pointing this out. Based on the comments raised by the Reviewer 2, some details of the methods used to detect the physiologic indexes have been added in the revised manuscript, which might further contribute to a too-long text. Therefore, to avoid this problem you proposed, we have rearranged Table 1, Figure 4, and Figure 5, as well as Figure 8 (before revision) to the attached supplementary files, and renamed Table S2, Figure S1, Figure S2, and Figure S4 based on the organization of the manuscript, respectively. Accordingly, the corresponding annotations in the original paper have been updated. Moreover, based on your comment on the caption ‘3.3 Analysis of cis‑acting regulatory elements in the promoters of the GH3 gene family in pear’, some unnecessary results have been removed (Lines 521−541). Please find the details in the revised manuscript.
Comment 4: Keywords cannot repeat the words from the title
Responses: Thank you for your comment. The ‘Keywords’ of this manuscript have been changed into the words as follows: Pear; GH3 gene family; PbrGH3.5; Fe deficiency hypersensitivity; reactive oxygen species; IAA.
Comment 5: The introduction must introduce the general functional traits researched here.
Responses: Thank you for your good comment. The related content has been added in Lines 136−150 (see below), and the associated references have been supplemented in the ‘References’. Meanwhile, based on the comments from the Reviewer 2, the ‘Introduction’ has been reorganized to make the content clearer and more logical. Please find the details in the ‘Introduction’.
Related content:
ROS are signaling molecules with dual effects in plants, mainly in the form of peroxide (H2O2), and superoxide radical (O2−) [42]. Mounting evidence has shown that Fe deficiency is closely associated with ROS. It was said that Fe deficiency would trigger the over-production of ROS, and cause irretrievable membrane damage to plants, hence leading to Fe deficiency hypersensitivity [43][44]. Therefore, to endure the Fe deficiency stress, the capacity to remove the excessive ROS is fundamental for plants, where superoxide dismutase (SOD) and peroxidase (POD) activities were of great significance [45]. According to the reports, plants with higher tolerance to Fe deficiency were commonly considered to be equipped with increased SOD and POD activities, and lower ROS accumulation under Fe deficiency stress [46][47]. Nevertheless, it remains to be explored whether the GH3-modulated FDR is associated with ROS.
Comment 6: The part of M&M must be organized, and statistics must be clear and clean.
Responses: We deeply appreciate the reviewer’s careful review. As you can find in the M&M of the revised manuscript, some details about the experimental design (Lines 224, 255, 303, 316, etc.), tissue collection (Lines 235, 245, 303, etc.), and plant age (Lines 292, 320, 340, etc.) have been explained, and some redundant content has been deleted (Lines 307−314). Besides, the statistical methods and analytical software applied in this paper have been listed in the text (Lines 271, 421, etc.). Please find the details in the main text.
Comment 7: The captions must be explained in detail! What is what in the table or Figures?
Responses: Thank you for your good comment. We have carefully read the captions of the Tables and Figures, and have made some improvements. The captions are listed below. Please find the details in the manuscript and supplementary materials.
Figure 1. Phylogenetic tree showing the relationships between GH3 gene families from pear, apple, and Arabidopsis.
Figure 2. Phylogenetic tree, conserved domain, and featured motifs of pear GH3 proteins (PbrGH3s).
Figure 3. Cis-acting regulatory elements (CREs) in the promoters of the GH3 genes in pear (PbrGH3s).
Figure 4. Ectopically overexpressed tomato plants with pear PbrGH3.5 are more sensitive to Fe deficiency stress.
Figure 5. Ectopic overexpression of pear PbrGH3.5 in tomato plants inhibits their capacities in Fe acquisition and ROS scavenging under Fe deficiency stress.
Figure 6. Prediction of the interaction and upstream regulation network of pear PbrGH3.5.
Figure S1. Expression heatmap of the GH3 gene family in pear.
Figure S2. A brief analysis of PbrGH3.5 in pear.
Figure S3. Molecular identification of PbrGH3.5-overexpressed transgenic tomato plants.
Figure S4. Ectopically overexpressing PbrGH3.5 in tomato plants facilitates the conjugation of IAA in their leaves under Fe-deficient conditions.
Table S1. The list of the GH3 genes used in blast analysis.
Table S2. Physicochemical characteristics of putative GH3 proteins in pear (PbrGH3s).
Table S3. The list of the primer pairs used in gene cloning and RT-qPCR analysis in pear and tomato plants.
Table S4. The promoter sequences of the GH3 gene family in the Chinese white pear (2000 bp upstream region of the corresponding genes was shown)
Comment 7: This last sentence of the ‘Introduction’ is not necessary, and you cannot judge.
Responses: Thank you for your suggestion. The sentence has been changed to ‘These findings established the foundation for further investigations on the PbrGH3s and shed light on the negative role of PbrGH3.5 in FDR’, which might be convincing.
Comment 8: How do you know that plants were in Fe deficiency?
Responses: Thank you for your question. Hydroponics using Hoagland nutrient solution without Fe is a common and effective way to induce Fe deficiency in Arabidopsis (Yan et al., 2016), tomatoes (Li et al., 2023), soybeans (Li et al., 2018), apples (Hao et al., 2022), etc. So, to obtain Fe deficiency-stressed pear plants, one-month-old ‘Duli’ pear seedlings were used, and subjected to such hydroponic conditions, and according to our unpublished results (showed below), the capacity of the roots rhizosphere acidification was notably activated in that situation (Figure 1a), and it was found that after exposure to Fe deficiency stress for 24 h, the SPAD value of the leaves was remarkedly decreased (Figure 1b). Moreover, the contents of Fe and Fe2+ were both reduced in pear leaves after 24 h of Fe deficiency treatment (Figure 1c, d), and the Fe content showed a statistical difference (Figure 1c). Collectively, we confirmed that after hydroponics for a period in this method, the plants are indeed in a state of Fe deficiency, which can be used for further study.
Figure 1. The obtaining of Fe deficiency-stressed pear seedlings. One-month-old soil-grown ‘Duli’ pear seedlings were subjected to hydroponic conditions (Hoagland nutrient solution without Fe), and (a) the capacity of the rhizosphere acidification of the roots, (b) the SPAD values of the leaves were monitored over time. (c) The contents of Fe, and (d) Fe2+ in leaves of pear seedlings exposed to Fe deficiency stress for 0 h and 24 h.
References:
Yan JY, Li CX, Sun L, Ren JY, Li GX, Ding ZJ, Zheng SJ. A WRKY Transcription Factor Regulates Fe Translocation under Fe Deficiency. Plant Physiol. 2016, 2017-2027.
Li X, Cao H, Yu D, Xu K, Zhang Y, Shangguan X, Zheng X, Yang Z, Li C, Pan X, Cui Y, Zhang Z, Han M, Zhang Y, Sun Q, Guo H, Zhao J, Li L, Li C. SlbHLH152, a bHLH transcription factor positively regulates iron homeostasis in tomato. Plant Sci. 2023, 335: 111821.
Li L, Gao W, Peng Q, Zhou B, Kong Q, Ying Y, Shou H. Two soybean bHLH factors regulate response to iron deficiency. J Integr Plant Biol. 2018, 60(7): 608-622.
Hao P, Lv X, Fu M, Xu Z, Tian J, Wang Y, Zhang X, Xu X, Wu T, Han Z. Long-distance mobile mRNA CAX3 modulates iron uptake and zinc compartmentalization. EMBO Rep. 2022, 23(5): e53698.
Comment 9: Results must be direct. Always indicate where we have to look, at the beginning of the paragraph, and please, describe all details in captions.
Responses: Thank you for your comment. As we have stated in responses to comment 6, all the captions of the Tables and Figures in the paper have been corrected to make the information clearer. However, we are sorry that we have not highlighted the corresponding Figures presenting the results at the beginning of the paragraph. Instead, we have added the annotations at the end of each sentence to allow readers to navigate the results. Please find the details in the ‘Results’
Comment 10: Discussion must start with YOUR NOVELTY.
Responses: Thank you for the kind reminder. The layout of the ‘Discussion’ has been reorganized with a focus on our findings, followed by a comparative analysis with other available reports, and finally conclusions. Please find the details in the ‘Discussion’ of the revised manuscript.
Comment 11: Please, improve the reference citations, this is one strongly perturbing element.
Responses: We are sorry for our negligence, and have corrected all the format of the citations. Please find the details in the main text.
We appreciate you for your warm work earnestly and hope that the changes we have made resolve all your concerns about the article. I’m more than happy to make any further changes that will improve the paper and/or facilitate successful publication.
Looking forward to hearing from you.
Yours Sincerely.
Bing Jia
jb1977@ahau.edu.cn

Reviewer 2 Report
Comments and Suggestions for Authors
The paper devoted to analysis of GH3 gene family in pear and it role in Fe response in tomato.
Authors do a hard work, but in the current from pare is not realy readable. From the formal point, authors must put all citations in the bracket like this [44], but not fused it with text body.
Title, abstracts, M&M require more clarity and logic, without “jumping” form one subject to other.
I would suggest to re-write carefully text, add all punctuation (brackets, comma etc) and made text more clear with further re-submission.
Title is very confusing. „Genome-wide analysis of the GH3 gene family in white pear (Pyrus bretschneideri) reveals the negative function of PbrGH3.5 in iron response.“ ?? How readers can know that Fe deficiency was studied in tomato?
Abstracts: need to be re-written with clear explanation of all results in simple form. In the current version only authors may know that experiments with iron was done in tomato.
M&M is not very clear.
Line 178: can you provide evidence that MicroTom required N-P-Cl ratio 48-1-4.4 what mean Cl 4 times more important as phosphate. Where I can read this?
Line 182: „12,000 lux“ = …. µmol/m2/sec
Line 205: „involved in Fe acquisition 45“ 45 genes???
Line 227: „ImagingWin software 49.“ ?? is 49 software?
Lines 202 and 228 – redundant.
Line 237: it is not a tomato, confusing for reader.
Figure 7, E,F,G H are unclear. Which SOD do you mean? Fe-SOD , Cu/Zn SOD or Mn SOD? Which POD? What do you mean supeoxide concentration? This panel may look as a main funding of the papre, what is not menation in Abstracts and title. I expact that you postulated that supeoxide in your hand extend half-life time significantly which can be considered as very new in chemistry. Right?
Plesae, provide all details of measure H2O2 and superoxide.
Line 521: „H2O2 and O2- levels“ ¿?? What is O2- level? O2- have a half-life less at 10 msec. So, level can not be measured. For the H2O2 the site of the production is a key, not general level in plants. H2O2 can be in peroxisomes with catalase as scavenging enzyme, in chloroplasts (stroma/thylakoid APX), cytoplalsm (cAPX). Effcet of each is unique and can not be combine in one value.
Figure 8 does not have too much sense: in the leaf IAA produced by different YUCCA-like gene (2011) and each cell type have own IAA contents. Fe deficiency caused dis-balance between IAA contents in different cell type and lead to cell-developmnet conflicts what resulted in growth and function inhibition. For exmpale, high IAA in mesophyll led to faster cell growth and tension pressure on epidermis cell etc. Clarification are required!
Comments on the Quality of English LanguageLayout need to be improved
Author Response
Responses to the reviewer 2
Dear editors and peer reviewers,
On behalf of my co-authors, we would like to thank you for giving us the opportunity to revise our manuscript. We appreciate the editor and the reviewers for the constructive comments on our manuscript entitled “Genome-wide analysis of the GH3 gene family in white pear (Pyrus bretschneideri) reveals the negative function of PbrGH3.5 in Fe deficiency response” (ijms-3311849). Those comments are valuable and helpful for reversing our paper, as well as important to improve our research work.
All the questions proposed by the reviews have been considered carefully and the manuscript has been revised accordingly. The reversion portion is marked in red on the paper. Our “Response to the reviewers’ Comments” were provided in blue fonts in the following text.
Reviewer 2,
Comment 1: The paper is devoted to the analysis of the GH3 gene family in pears and its role in Fe response in tomatoes. The authors do a hard job, but the current form of the paper is not readable. From the formal point, authors must put all citations in the bracket like this [44], but not fuse it with the text body.
Responses: Thank you for pointing this out, and we must apologize for our negligence. We have carefully checked the citation format of the references and presented them in the style you suggested and required by the journal. Please find the details in the revised manuscript.
Comment 2: Title, abstracts, and M&M require more clarity and logic, without “jumping” from one subject to another. I would suggest to re-write carefully, adding all punctuation (brackets, commas, etc.) and making the text more clear.
Responses: We sincerely appreciate the valuable suggestions. According to your comments, we have partially revised the ‘Introduction’ without changing the original structure and have added some content about the role of reactive oxygen species (ROS) in Fe deficiency responses (FDR) given the comment of the Reviewer 1. And based on your suggestion on the writing of the ‘Abstract’, we have adjusted the last paragraph to emphasize the findings and focus of this paper. Please find the details in the ‘Introduction’ of the revised manuscript.
Comment 3: The title is very confusing. How readers can know that Fe deficiency was studied in tomatoes?
Responses: Thank you for your kind reminder. Based on your comment, we have adjusted the title to ‘Identification of the GH3 gene family in Chinese white pear (Pyrus bretschneideri) and functional analysis of PbrGH3.5 in Fe deficiency response in tomato’. Please find it in the revised manuscript.
Comment 4: Abstracts: need to be re-written with a clear explanation of all results in simple form. In the current version authors only may know that experiments with iron were done in tomatoes.
Responses: Thank you for your constructive comment. We agree that the abstract should be reorganized to reflect the overall content and results of the manuscript. According to the suggestion, a brief but clear description of the main findings of this manuscript was added. Please find the details in Lines 16−56 in the revised manuscript with track changes.
Comment 5: M&M is not very clear.
Responses: Thank you for your kind reminder. According to the comment, some detail about the experiment was supplemented in the M&M, including the brief introduction of the tested tissues (Lines 226−246), the procedure to mimic Fe deficiency stress of pear seedlings (Lines 313−324), the methods used for Agrobacterium-mediated leaf disk transformation of tomato plants (Lines 337−346), the details for the determination of chlorophyll (Lines 356−363), relative electrolyte leakage (REL, Lines 368−375), and ferric chelate reductase (FCR) activity (Lines 390−3995), as well as the presentations of software used for column charts construction and analysis of significant differences (Lines 417−419), etc. Please find the details in the revised manuscript.
Comment 6: Line 178, can you provide evidence that MicroTom required an N-P-Cl ratio of 48-1-4.4 which means Cl is 4 times more important than phosphate? Where I can read this?
Responses: Thanks for your questions. However, we’re confused about this content. After carefully checking the details, we haven’t found the related content in Line 178 in the original manuscript (see below). Thus, we wondered whether it was a misunderstanding of the related content in the original manuscript caused by the format of the citation, just as you have stated in Comment 7, for which, we must express our sincere apologies. Of course, we might have missed this statement in the revision process, if so, we would appreciate the feedback from you.
The related content in Line 178 before revision:
Comment 7: Line 182, lux = µmol/m2/sec; Line 205, what’s the meaning of the number ‘45’? 45 genes? Line 227, what’s the meaning of the number ‘49’? Software?
Responses: Thanks for your careful review. According to your suggestion, the ‘lux’ in Line 182 (before revision) has been changed in the form of µmol∙m-2∙sec-1. (Line 282 in the revised manuscript); As for the numbers ‘45’ in Line 205 and ‘49’ in Line 227, they referred to the corresponding order of the cited reference, and based on your Comment 1, we have corrected all the citation format to avoid unwanted misreading. Please find the details in the revised manuscript.
Comment 8: Lines 202 and 228 – redundant.
Responses: Thank you for pointing this out, and we are sorry for the redundant content. As you can see from the revised manuscript, the corresponding content in Line 202 (before revision) has been deleted due to the adjustment of the writing style (Lines 304−310), and the related content has been added in Lines 381−387 of the revised manuscript. Besides, based on your Comment 5, the catalogs of the commercially available kits were added. Please find it in the revised manuscript.
Comment 9: Line 237: it is not a tomato, confusing for the reader.
Responses: Thank you for your kind reminder. To avoid confusion among readers, the subheading in Line 237 has been changed to ‘qRT-PCR analysis’. Likewise, some improper writing has been corrected as well, such as Lines 73, 466, 633, 653, 751, etc. Please find the details in the revised manuscript. Please find the details in the revised manuscript.
Comment 10: The contents of Figure 7E−H are unclear. Which SOD do you mean? Fe-SOD, Cu/Zn SOD or Mn SOD? Which POD? Guaiacol-POD? Glutathione-POD, or ascorbic acid-POD? What do you mean by superoxide concentration? This panel may look like a main finding of the paper, which is not mentioned in the Abstracts and title.
Responses: We deeply appreciate the reviewer’s careful review. Following your reminders, we have consulted with the technical staff about the detail of the commercial kit (BC5165, https://www.solarbio.com/) that is widely used to measure the SOD activity in plant tissues, and have confirmed that the SOD examined in this paper includes all the forms you mentioned.
However, in the case of POD, it specifically refers to guaiacol-POD since it is detected with guaiacol as the substrate. Considering that these indicators were quantified using the commercially available kits, we did not add details about the procedures. Instead, we supplied the EC number of SOD and POD (see Line 383);.
As for the ‘superoxide concentration’ (Line 700, before revision), we recognized that it was an unflattering presentation, and deleted it based on the comment (proposed by the Reviewer 1) that the structure of the ‘Discussion’ should be reorganized, which you could find in Lines 898−925; finally, as you have stated that the related results are the main funding of our study, hence, according to your Comment 4, these findings have been presented in the ‘Abstract’. Please find the details in the revised manuscript.
Comment 11: Please, provide all details of measuring H2O2 and superoxide (SOD).
Responses: Thank you for your comment. Given the following two points, we are sorry that this comment was not adopted. Firstly, as we have emphasized in the manuscript, the detections of H2O2 level and SOD activity were all completed using the commercial kits, but the not traditional method with or without some modification; secondly, given the comment of the Reviewer 1, the content of this manuscript seemed to be overloaded and needed to be streamlined. Thereby, to avoid this issue, the Tables and Figures have been rearranged, and provided as supplements. Please find the revised manuscript and supplementary materials.
Comment 12: Line 521: H2O2 and O2- levels, what is O2- level? O2- has a half-life of less than 10 msec. So, the level can not be measured. For the H2O2 the site of the production is a key, not a general level in plants. H2O2 can be in peroxisomes with catalase as a scavenging enzyme, in chloroplasts (stroma/thylakoid APX), cytoplasm (cAPX). The effect of each is unique and can not be combined in one value.
Responses: Thank you for your good comment. From the questions you presented, you must be an expert who is skilled in ROS and has obtained great achievements, which are beyond our research scope so, we learned a lot. According to the instructions of the related kit, the level of O2- could be indirectly detected. In that case, the sample was grinded using the extraction solution and subsequently centrifuged at 4℃ to collect the supernatant. Then, the supernatant was reacted with hydroxylamine hydrochloride to form NO2-, and the resulting NO2- was immediately conjugated under the actions of sulfanilic acid and α-Naphthylamine. Finally, the absorbance of the generated red conjugates was detected at 530 nm, and the value was utilized to quantify the O2- level. Notably, to avoid excessive content of the paper, the detail to measure the O2- level was ignored.
Just as done for O2- detection, the whole leave sample was grinded using the extraction solution and subsequently centrifuged at 4℃ to collect the supernatant, which was subsequently used for analysis. Therefore, we considered that H2O2 determined in this study indicated the H2O2 in all tissues and organelles, which was confirmed by the technical staff after consultation. Although various H2O2 might be unique and the combined H2O2 level may not be appropriate to evaluate its function in some specific biological progress, it might at least provide a critical referential significance in evaluating plant resistances to abiotic stresses (Song et al., 2022; Ma et al., 2024; Zhao et al., 2024), including Fe deficiency stress (Li et al., 2021; Zhai et al., 2021).
References:
Song J, Sun P, Kong W, Xie Z, Li C, Liu JH. SnRK2.4-mediated phosphorylation of ABF2 regulates ARGININE DECARBOXYLASE expression and putrescine accumulation under drought stress. New Phytol. 2023, 238(1): 216-236.
Ma G, Liu Z, Song S, Gao J, Liao S, Cao S, Xie Y, Cao L, Hu L, Jing H, Chen L. The LpHsfA2-molecular module confers thermotolerance via fine-tuning of its transcription in perennial ryegrass (Lolium perenne L.). J Integr Plant Biol. 2024. doi: 10.1111/jipb.13789.
Zhao H, Jia Y, Niu Y, Wang Y. The BpPP2C-BpMADS11-BpERF61 signaling confers drought tolerance in Betula platyphylla. New Phytol. 2024, 244(6): 2364-2381.
Li D, Sun Q, Zhang G, Zhai L, Li K, Feng Y, Wu T, Zhang X, Xu X, Wang Y, Han Z. MxMPK6-2-bHLH104 interaction is involved in reactive oxygen species signaling in response to iron deficiency in apple rootstock. J Exp Bot. 2021, 72(5): 1919-1932.
Zhai L, Sun C, Li K, Sun Q, Gao M, Wu T, Zhang X, Xu X, Wang Y, Han Z. MxRop1-MxrbohD1 interaction mediates ROS signaling in response to iron deficiency in the woody plant Malus xiaojinensis. Plant Sci. 2021, 313: 111071.
Comment 13: Figure 8 does not make too much sense: in the leaf, IAA is produced by different YUCCA-like genes (2011) and each cell type has its own IAA contents. Fe deficiency caused dis-balance between IAA contents in different cell types and led to cell-development conflicts which resulted in growth and function inhibition. For example, high IAA in mesophyll led to faster cell growth and tension pressure on epidermis cells, etc. Clarification is required!
Responses: Thank you for your comment. It is reported that the GH3 members in Group Ⅱ have IAA-amido synthetases, and could affect the homeostasis of free IAA and conjugated IAA. Based on the phenotypic tree of PbrGH3s, we found that PbrGH3.5 belonged to Group Ⅱ, which implies its role in conjugating free IAA. Given that IAA is a kind of vital and positive hormone in enhancing the Fe deficiency tolerance of plants, we, therefore, wonder whether the impaired Fe deficiency tolerance of the PbrGH3.5-overexpressed transgenic tomato plants was partially associated with IAA in it. To support the hypothesis, the contents of the free IAA and various IAA conjugations were measured, and the results were shown and discussed, which was a key part of this work. However, as you have claimed, each cell type has its own IAA content and may result in conflicts in growth and function. Hence, combining your suggestion and our consideration, we have deleted Figure 8 in the main text and instead presented it as Figure S4. Please find the details in the manuscript and supplementary materials.
Comment 14: Layout needs to be improved
Responses: Thank you for your kind reminder. The layout of the manuscript has been restructured to a greater extent. Please find the details in the revised manuscript.
We appreciate you for your warm work earnestly and hope that the changes we have made resolve all your concerns about the article. I’m more than happy to make any further changes that will improve the paper and/or facilitate successful publication.
Looking forward to hearing from you.
Yours Sincerely.
Bing Jia
jb1977@ahau.edu.cn

Round 2
Reviewer 1 Report
Comments and Suggestions for Authors
Title – improved
Abstract - improved
Keywords - organize keywords in alphabetic order
Introduction - improved
M&M - all software must have a reference or link.
Try to pool such repetitive phrases in one (lines 229 and 241).
Statistical analyses - Please, explain the FACTOR and the ANALYZED number of levels.
Table S3 was mentioned in the text before the Table S2. Change the order.
Results – never use the references and discussion in RESULTS, as are lines 658-665; 671-676; 702-703.
Details are done in pdf file

Author Response
Responses to the reviewer 1
Dear editors and peer reviewers,
On behalf of my co-authors, we would like to thank you for giving us the opportunity to revise our manuscript. We appreciate the editor and the reviewers for the constructive comments on our manuscript entitled “Identification of the GH3 gene family in Chinese white pear (Pyrus bretschneideri) and functional analysis of PbrGH3.5 in Fe deficiency responses in tomato” (ijms-3311849). Those comments are valuable and helpful for reversing our paper, as well as important to improve our research work.
All the questions proposed by the reviews have been considered carefully and the manuscript has been revised accordingly. The reversion portion is marked in red on the paper. Our “Response to the reviewers’ Comments” were provided in blue fonts in the following text.
Reviewer 1
Comment 1: Title – improved
Response: Thank you for your time and effort in reviewing the manuscript.
Comment 2: Abstract – improved
Response: Thank you for your time and effort in reviewing the manuscript.
Comment 3: Keywords - organize keywords in alphabetic order
Response: Thank you for your suggestion. The Keywords have been reorganized in alphabetic order (Lines 45, 46). Please find the details in the revised manuscript.
Comment 4: Introduction – improved
Response: Thank you for your time and effort in reviewing the manuscript.
Comment 5: M&M - all software must have a reference or link.
Response: Thank you for your kind reminder. The missed references or links have been added to the main text. Please see them in Lines 163, 217, 193, 242, and 379.
Comment 6: Try to pool such repetitive phrases in one (lines 229 and 241).
Response: Thank you for your kind reminder. The redundant phrases have been combined and presented at the end of the paragraph. Please see it in Line 261.
Comment 7: Statistical analyses - Please, explain the factor and the analyzed number of levels.
Response: Thank you for your comment. The revised ‘Statistical analysis’ is as follows:
All the experiments were carried out in three independent biological replicates. GraphPad Prism 9.5 software and Microsoft Excel software 2017 were applied to construct the heatmaps and column charts. Data were shown as mean ± standard deviation (SD). The SPSS v 26.0 software (SPSS Inc., Chicago, IL, USA) was applied to analyze the statistical differences using one-way analysis of variance (ANOVA) with Duncan’s multiple range test or Student’s t-test, and the significant differences were indicated by different lowercases at the level of p < 0.05.
Comment 8: Table S3 was mentioned in the text before the Table S2. Change the order.
Response: Thank you for your kind reminder. The order has been revised. Please find it in the manuscript and the supplementary materials.
Comment 9: Results – never use the references and discussion in RESULTS, as are lines 658-665; 671-676; 702-703.
Response: Thank you for your constructive comment. The sentence in the corresponding position has been deleted or changed. In Lines 658-665, the sentence has been changed to ‘To explore the potential working pathways of PbrGH3.5, the expressions of the Fe uptake-related genes in tomato plants were examined’; In Lines 671-676, the sentence has been deleted; In Lines 702-703, the sentence has been replaced with the content ‘To investigate that relationship between the PbrGH3.5-diminished FDR with IAA’. Please find the details in the manuscript.
Comment 10: Some details marked in the PDF file should be noted and be corrected.
Response: Thank you for your careful review. Some mistakes have been corrected, please find the details in the manuscript.
We appreciate you for your warm work earnestly and hope that the changes we have made resolve all your concerns about the article. I’m more than happy to make any further changes that will improve the paper and/or facilitate successful publication.
Looking forward to hearing from you.
Yours Sincerely.
Bing Jia
jb1977@ahau.edu.cn

Reviewer 2 Report
Comments and Suggestions for Authors
Thank you! The text is better, but some major point need to be corrected.
Line 280: you mentioned here MS salt (not MS medium, since original MS contain hormones). If look for ion balance you apply for MictoTom (N:P:Cl) you will find ratio 44:1:4,8) what automatically mean that in your case chloride have a some biological function and more important as phosphate. Please, be carefully and dont apply such nutritional stress to plants.
Comment 12: „According to the instructions of the related kit, the level of O2- could be indirectly detected. In that case, the sample was grinded using the extraction solution and subsequently centrifuged at 4℃ to collect the supernatant. Then, the supernatant was reacted with hydroxylamine hydrochloride to form NO2-, and the resulting NO2- was immediately conjugated under the actions of sulfanilic acid and α-Naphthylamine. Finally, the absorbance of the generated red conjugates was detected at 530 nm, and the value was utilized to quantify the O2- level. Notably, to avoid excessive content of the paper, the detail to measure the O2- level was ignored.” – I think it can be the manin finding of the paper- extension of superoxide half-life to hours. No one do this before. Please, add more details how can you do this. Before your protocol superoxde half-life is less as 10 millisec only.
Line 640: “Ectopically overexpressed tomato plants with pear PbrGH3.” – Plants can noe be overexpressed, gene can be. Please, changes sentence structure.
Line 673: Damage is not a direct effect of the ROS accumulation. The most dangerous is conflicts caused by local ROS. Different cell type differntly respond to ROS and caused constrains between cell type. This may led to damage. Please, consider it in future.
Lines 715 – 717: edition required.
IAA measure: it is the same as ROS: tomato have many IAA production gene which regulated differentially and im-balance between it is a reason of growth inigibtion. https://www.sciencedirect.com/science/article/pii/S0981942811000787
Total IAA have a less importance. Please, mention this and your ideas.
Please, if possible, load "clean" version of the paper. In the current one it is not so easy torecognise final version.
Author Response
Responses to the reviewer 2
Dear editors and peer reviewers,
On behalf of my co-authors, we would like to thank you for giving us the opportunity to revise our manuscript. We appreciate the editor and the reviewers for the constructive comments on our manuscript entitled “Identification of the GH3 gene family in Chinese white pear (Pyrus bretschneideri) and functional analysis of PbrGH3.5 in Fe deficiency responses in tomato” (ijms-3311849). Those comments are valuable and helpful for reversing our paper, as well as important to improve our research work.
All the questions proposed by the reviews have been considered carefully and the manuscript has been revised accordingly. The reversion portion is marked in red on the paper. Our “Response to the reviewers’ Comments” were provided in blue fonts in the following text.
Reviewer 2
Comment 1: Thank you! The text is better, but some major points need to be corrected. In Line 280, you mentioned here MS salt (not MS medium, since original MS contain hormones). If look for ion balance you apply for MicroTom (N:P:Cl) you will find ratio 44:1:4,8) what automatically mean that in your case chloride have a some biological function and more important as phosphate. Please, be carefully and don’t apply such nutritional stress to plants.
Response: Thank you for your suggestions. The clerical error has been corrected into ‘MS base salt’ in Line 251. According to the feedback from the technical staff, the MS base salt (Table 1) is prepared under the accepted standard, which would not cause nutritional stress to tomato plants and is suitable for tissue-culture of tomato seedlings (Park et al., 2003; Chetty et al., 2013; Guo et al., 2024). I knew your concern that chloride in the MS salt media might impair some unknown biological progresses that may be harmful to tomato seedling, but as had been stated in published reports (Guo et al., 2013; Li et al., 2013), the seedling cultured in MS salt media grew well, and did not display any unpredicted abnormal phenomenon, such as chlorosis. Importantly, the tissue-cultured tomato seedings were only used for genetic transformation (Line 255) in our study, and the soil-grown tomato plants were subjected to Fe deficiency stress (Line 261). These details might point out the chloride could be principally ignored, and strongly support the findings of our study is independent on chloride. Of course, considering your kind reminder, we will pay more attention to the experiment design in our future works, thanks a lot.
Table 1 The reagent formulation of MS base salt
References:
Park, S.H., Morris, J., Park, J.E., Hirschi, K.D., Smith, R.H. Efficient and genotype-independent Agrobacterium-mediated tomato transformation. Journal of Plant Physiology, 2003, 160, 1253–1257.
Chetty, V.J., Ceballos, N., Garcia, D., Narvaez‐Vasquez, J., Lopez, W., Orozco‐Cardenas, M.L. Evaluation of four Agrobacterium tumefaciens strains for the genetic transformation of tomato (Solanum lycopersicum L.) cultivar Micro‐Tom. Plant Cell Reports, 2013, 32, 239–247.
Guo, G.L., Wei, P.W., Yu, T., Zhang, H.Y., Heng, W., Liu, L., Zhu, L.W., Jia, B. PbrARF4 contributes to calyx shedding of fruitlets in ‘Dangshan Suli’ pear by partly regulating the expression of abscission genes. Horticultural Plant Journal, 2024, 10, 341–35.4
Guo, G.L., Yu, T., Zhang, H.Y., Chen, M., Dong, W., Zhang, S.Q., Tang, X.M., Liu, L., Heng, W., Zhu, L.W., Jia, B. Evidence that PbrSAUR72 contributes to iron deficiency tolerance in pears by facilitating iron absorption. Plants-Basel 2023, 12, 2173.
Comment 2: ‘According to the instructions of the related kit, the level of O2- could be indirectly detected. In that case, the sample was grinded using the extraction solution and subsequently centrifuged at 4℃ to collect the supernatant. Then, the supernatant was reacted with hydroxylamine hydrochloride to form NO2-, and the resulting NO2- was immediately conjugated under the actions of sulfanilic acid and α Naphthylamine. Finally, the absorbance of the generated red conjugates was detected at 530 nm, and the value was utilized to quantify the O2- level. Notably, to avoid excessive content of the paper, the detail to measure the O2- level was ignored.’ – I think it can be the main finding of the paper- extension of superoxide half-life to hours. No one do this before. Please, add more details how can you do this. Before your protocol superoxde half-life is less as 10 millisec only.
Response: Thank you for your comment, and we are very sorry for not being able to provide more details, which is really a challenge to us. As we have stated, the O2- level was indirectly detected using commercial obtained kits from Solarbio (Beijing, China), and the detailed instruction to conduct this experiment is provided in our previous responses. Indeed, the corresponding kits is not only available from Solarbio (Jia et al., 2024), but also from Jiancheng (Lin et al., 2024; Nanjing, China), Geruisi (Zhu et al., 2024; Suzhou, China), Comin (Mao et al., 2023; Suzhou, China), etc. and the indictor has been widely applied to evaluate the responses of plants under abiotic stresses in these aforementioned studies. We must admit that we are really not familiar with the concept of the half-life for O2-, and we are not sure about the diffcultities in measuring the level for it, which should attract our attention later. However, based on these aforementioned reports, we believe that the O2- level in plants is quantifiable an the kit assay in our study is feasible and informative.
References:
Mao K, Yang J, Sun Y, Guo X, Qiu L, Mei Q, Li N, Ma F. MdbHLH160 is stabilized via reduced MdBT2-mediated degradation to promote MdSOD1 and MdDREB2A-like expression for apple drought tolerance. Plant Physiol. 2024, 194(2): 1181-1203.
Jia Z, Zeng T, Gu L, Wang H, Zhu B, Ren M, Du X. TaWRKY17 interacts with TaWRKY44 to promote expression of TaDHN7 for salt tolerance in Wheat. Plant Cell Environ. 2024. doi: 10.1111/pce.15277.
Lin L, Yuan K, Qi K, Xie Z, Huang X, Zhang S. Synergistic interaction between PbbZIP88 and PbSRK2E enhances drought resistance in pear through regulation of PbATL18 expression and stomatal closure. Plant Cell Environ. 2024. doi: 10.1111/pce.15131.
Zhu C, Lin Z, Liu Y, Li H, Di X, Li T, Wang J, Gao Z. A bamboo bHLH transcription factor PeRHL4 has dual functions in enhancing drought and phosphorus starvation tolerance. Plant Cell Environ. 2024, 47(8): 3015-3029.
Comment 3: Line 640: “Ectopically overexpressed tomato plants with pear PbrGH3.” – Plants can not be overexpressed, gene can be. Please, changes sentence structure.
Response: Thank you for your kind reminder. The sentence has been revised to ‘PbrGH3.5-overexpressed tomato plants are more sensitive to Fe deficiency stress.’. Please find the details in Line 568.
Comment 4: Line 673: Damage is not a direct effect of the ROS accumulation. The most dangerous is conflicts caused by local ROS. Different cell types differently respond to ROS and caused constrains between cell type. This may lead to damage. Please, consider it in future.
Response: Thank you for your reminder and constructive comment. We have learned a lot form you, and will be careful in our future study.
Comment 5: Lines 715 – 717: edition required.
Response: Thank you for your careful review. The missed words have been added, and the corrected sentence is as follows: These findings illustrate that the decline in IAA among PbrGH3.5-overexpressed tomato plants may account for their Fe deficiency sensitivity.
Comment 6: IAA measure: it is the same as ROS: tomato have many IAA production gene which regulated differentially and im-balance between it is a reason of growth inigibtion. https://www.sciencedirect.com/science/article/pii/S0981942811000787. Total IAA have a less importance. Please mention this and your ideas.
Response: Thank you for your suggestion. Given the viewpoint you proposed, we have added related content in the ‘Discussion’ (Lines 809−814), and attached the corresponding citation in the ‘References’. Please find the details in the revised manuscript.
Comment 7: Please, if possible, load "clean" version of the paper. In the current one it is not so easy to recognise final version.
Response: We are sorry for your trouble in reading and reviewing. The revised manuscript is available in a clean version, and the changes we have made are presented in yellow background with corresponding annotations. Please find the details in the revised manuscript.
We appreciate you for your warm work earnestly and hope that the changes we have made resolve all your concerns about the article. I’m more than happy to make any further changes that will improve the paper and/or facilitate successful publication.
Looking forward to hearing from you.
Yours Sincerely.
Bing Jia
jb1977@ahau.edu.cn

Round 3
Reviewer 1 Report
Comments and Suggestions for Authors
The basic problems are still in the fact that:
1. No gap and hypotheses were established at the end of the Introduction, but one extremely arrogant exposition of results and accomplishments and importance for the world science. Bad structure and bad mode of writing.
2. Statistical analyses - Please explain, you have 8 levels, e for ? and 4 for what? What was the factor, which levels were? Please define, this is the third time that I am asking
3. Not having hypotheses, your discussion is errant and not very agreeable to read, and the Conclusions are simple repetition of results. So, if you make a correct structure, you will improve the quality of your presentation. And of clearness.
Details are done in pdf file

Author Response
Responses to the reviewer 1
Dear editors and peer reviewers,
On behalf of my co-authors, we would like to thank you for giving us the opportunity to revise our manuscript. We appreciate the editor and the reviewers for the constructive comments on our manuscript entitled “Identification of the GH3 gene family in Chinese white pear (Pyrus bretschneideri) and functional analysis of PbrGH3.5 in Fe deficiency responses in tomato” (ijms-3311849). Those comments are valuable and helpful for reversing our paper, as well as important to improve our research work.
All the questions proposed by the reviews have been considered carefully and the manuscript has been revised accordingly. The reversion portion is marked in red on the paper. Our “Response to the reviewers’ Comments” were provided in blue fonts in the following text.
Reviewer 1
Comment 1: No gap and hypotheses were established at the end of the Introduction, but one extremely arrogant exposition of results and accomplishments and importance for the world science.
Response: Thank you for your constructive comment. The revised paragraph is as follows, and the gap and hypotheses were addressed. Please finds the detail in the ‘Introduction’ of the revised manuscript.
Revised content:
While it has been more than a decade since the release of the first genome of pear [48], the knowledge of the pear GH3 gene family (PbrGH3s) is absent, and whether and how PbrGH3s are engaged in the FDR of plants have not been explored. Here, the GH3 gene family from pear (Pyrus bretschneideri) was isolated to explore the family members, gene structure, conserved motifs, phylogenetic evolution, cis-acting regulatory elements, tissue-specific expression properties, and their expression patterns in the normal (NL) and Fe-deficient chlorotic leaves (CL) of pear trees. Combined with published RNA-seq data and qRT-PCR analysis, we found that PbrGH3.5 was strongly induced by Fe deficiency. We then hypothesized that PbrGH3.5 might participate in the response to Fe deficiency. Subsequently, the function of PbrGH3.5 was investigated through a reverse-genetic approach., and the results showed that overexpression of PbrGH3.5 could reduce the Fe deficiency tolerance of tomato plants, which are model plants that share the same Fe absorption system with pear trees and are widely used for Fe deficiency analysis [49][50]. These findings unveiled crucial mechanisms of plant GH3-mediated responses to Fe deficiency, and provided valuable information and a theoretical framework for the functional investigation of pear GH3 gene members.
Comment 2: Statistical analyses - Please explain, you have 8 levels, e for? and 4 for what? What was the factor, which levels were? Please define, this is the third time that I am asking.
Response: Thank you for your patience, and please accept our apologize for our carelessness and misunderstanding. The related contents have been complemented in the Figure legends of Fig. 4, Fig. 5, Fig. S2, and Fig. S3. The revised contents are as follows.
Revised content:
Figure 4. .... The data were shown as the mean ± standard deviation (SD) of three biological replicates (n = 3), and the statistical differences were indicated by different lowercases (a−c) or asterisks at p < 0.05 (one-way ANOVA with Duncan’s multiple range test for charts b, e, d, f, and h, and Student’s t-test for chart i).
Figure 5. ... The data were shown as the mean ± standard deviation (SD) of three biologically independent (n = 3), and the statistical differences were indicated by different lowercases (a−c) at p < 0.05 (one-way ANOVA with Duncan’s multiple range test).
Figure S2. ... The data were shown as the mean ± standard deviation (SD) of three independent replicates (n = 3), and the statistical differences were indicated by different lowercases (a−d) at p < 0.05 (one-way ANOVA with Duncan’s multiple range test).
Figure S3. ... The data were shown as the mean ± standard deviation (SD) of three biological replicates (n = 3), and the statistical differences were indicated by different lowercases (a−c) or asterisks at p < 0.05 (one-way ANOVA with Duncan’s multiple range test for charts a and b, and Student’s t-test for chart c)
Figure S4. ... The data were shown as the mean ± standard deviation (SD) of three replicates (n = 3), and the statistical differences were indicated by different lowercases (a, and b) at p < 0.05 (one-way ANOVA with Student’s t-test).
Comment 3: Not having hypotheses, your discussion is errant and not very agreeable to read, and the Conclusions are simple repetition of results. So, if you make a correct structure, you will improve the quality of your presentation. And of clearness.
Response: Thank you for valuable suggestions. The related content about hypotheses in the ‘Discussion’ have been thoroughly deleted or replaced by other words. Please find the details in Lines 712, 731, 761, 817, and 834. Based on your comment, the ‘Conclusion’ has been reorganized. The revised ‘Conclusion’ is provided below.
5 Conclusion
Collectively, a total of 15 GH3 gene members in the Chinese white pear (PbrGH3s) were examined and could be grouped into two subclusters. Most of these members are highly conserved in the gene structure and protein motifs, and showed tissue-specific expression preferences. PbrGH3s might function in physiological processes associated with hormone signaling and abiotic stress, and several PbrGH3s may participate in Fe deficiency responses. Among them, PbrGH3.5, a potential interactor of PbrSnRKs and putative target downstream of PbrDofs, was characterized to negatively regulate the Fe deficiency responses of tomato plants in ROS- and IAA-dependent manners. Overall, our results established the foundation for further comprehensive investigations of PbrGH3s.
Other changes based on the available PDF file:
1, The ‘keywords’ has been renewed in Lines 45 and 46;
2, ‘Beyond these’ in Line 133;
3, The title of 3.2 has been changed to ‘Phylogenetic evolution, gene structure and motif analysis of pear GH3 gene family’;
4, The sentence in Lines 118-120 (before revision) has been changed to ‘ROS are predominantly present in the form of peroxide (H2O2), and superoxide radical (O2−) in plants, and serve as signaling molecules with dual effects in regulating multiple biological processes’;
5, The sentence ‘It was said’ in Line 121 (before revision) has been deleted;
6, The font in Lines 393, 432, 656 (before revision) has been corrected.
Please find the details in the revised manuscript.
We appreciate you for your warm work earnestly and hope that the changes we have made resolve all your concerns about the article. I’m more than happy to make any further changes that will improve the paper and/or facilitate successful publication.
Looking forward to hearing from you.
Yours Sincerely.
Bing Jia
jb1977@ahau.edu.cn

Reviewer 2 Report
Comments and Suggestions for Authors
Thank you for the clear answer and providing last not-reliable citation about kit and superoxide measure. As I understood, that in your case superoxde can extend half-life (this is a very new statement for chemistry) to least 30 min (from 10msec) and still present in your supernatant. "Superoxide is considered both a radical and a −1 charged anion. It is a relatively unstable molecule, with a half-life of milliseconds, a reasonably strong oxidant, in which case it is reduced to hydrogen peroxide, and can also act as a reductant and convert to oxygen." . Which conditions for stabilization of superoxide and, from the oether side, avoid any SOD activity in your solution you have used?
My best regards!
Author Response
Responses to the reviewer 2
Dear editors and peer reviewers,
On behalf of my co-authors, we would like to thank you for giving us the opportunity to revise our manuscript. We appreciate the editor and the reviewers for the constructive comments on our manuscript entitled “Identification of the GH3 gene family in Chinese white pear (Pyrus bretschneideri) and functional analysis of PbrGH3.5 in Fe deficiency responses in tomato” (ijms-3311849). Those comments are valuable and helpful for reversing our paper, as well as important to improve our research work.
All the questions proposed by the reviews have been considered carefully and the manuscript has been revised accordingly. The reversion portion is marked in red on the paper. Our “Response to the reviewers’ Comments” were provided in blue fonts in the following text.
Reviewer 2
Comment: Thank you for the clear answer and providing last not-reliable citation about kit and superoxide measure. As I understood, that in your case superoxde can extend half-life (this is a very new statement for chemistry) to least 30 min (from 10 msec) and still present in your supernatant. "Superoxide is considered both a radical and a −1 charged anion. It is a relatively unstable molecule, with a half-life of milliseconds, a reasonably strong oxidant, in which case it is reduced to hydrogen peroxide, and can also act as a reductant and convert to oxygen.". Which conditions for stabilization of superoxide and, from the other side, avoid any SOD activity in your solution you have used.
Response: Thank you for your suggestion, and please accept our apologize for your confusion. Unfortunately, owing to our limited cognition and knowledge, we are unable to provide more details and other sufficient data or results to illustrate the issues you have raised. In any case, this is a key point that we should be aware of and pay attention to in our future study. Thank you for pointing this out.
We appreciate you for your warm work earnestly and hope that the changes we have made resolve all your concerns about the article. I’m more than happy to make any further changes that will improve the paper and/or facilitate successful publication.
Looking forward to hearing from you.
Yours Sincerely.
Bing Jia
jb1977@ahau.edu.cn
